# Sex-dependent and sex-independent regulatory systems of size variation in natural populations

Hirokazu Okada[1],[*] , Ryohei Yagi[1] , Vincent Gardeux[2] , Bart Deplancke[2] & Ernst Hafen[1],[3],[**]

## Abstract

Size of organs/organisms is a polygenic trait. Many of the growth-regulatory genes constitute conserved growth signaling pathways. However, how these multiple genes are orchestrated at the systems level to attain the natural variation in size including sexual size dimorphism is mostly unknown. Here we take a multi-layered systems omics approach to study size variation in the *Drosophila* wing. We show that expression levels of many critical growth regulators such as Wnt and TGFβ pathway components significantly differ between sexes but not between lines exhibiting size differences within each sex, suggesting a primary role of these regulators in sexual size dimorphism. Only a few growth genes including a receptor of steroid hormone ecdysone exhibit association with between-line size differences. In contrast, we find that between-line size variation is largely regulated by genes with a diverse range of cellular functions, most of which have never been implicated in growth. In addition, we show that expression quantitative trait loci (eQTLs) linked to these novel growth regulators accurately predict population-wide, between-line wing size variation. In summary, our study unveils differential gene regulatory systems that control wing size variation between and within sexes.

**Keywords** growth; omics; sexual dimorphism; size; wing
**Subject Categories** Development; Evolution & Ecology; Genetics, Gene Therapy & Genetic Disease
**Mol Syst Biol.** (2019) 15: e9012

## Introduction

The mechanisms that control the size of cells, organs, and organisms have long been a fundamental theme in biology (Oldham *et al*, 2000; Edgar, 2006; Andersen *et al*, 2013). Over the past decades, forward and reverse genetic approaches in model organisms including *Drosophila* have identified many genes that control growth and size. These analyses successfully identified several conserved growth-regulatory signaling pathways such as Wnt, transforming growth factor beta (TGFβ), Hippo, and the mechanistic target of rapamycin (mTOR), whose deregulation has been linked to cancers (Massagué *et al*, 2000; Logan & Nusse, 2004; Laplante & Sabatini, 2012; Tumaneng *et al*, 2012; Misra & Irvine, 2018). These single gene approaches, however, were unable to evaluate how those multiple genes/pathways interact to determine size. It is therefore still unclear whether the conserved growth pathways are indeed the critical determinants of size variation observed in natural populations. If so, how then are the pathway components controlled genetically to cause the observed size variation? Sexual size dimorphism is common throughout the animal kingdom, yet the molecular mechanism remains poorly understood (Williams & Carroll, 2009). In *Drosophila*, females have a larger body and wings than males. The distinct number of X chromosomes (2 in females, 1 in males) causes differential activation of *Sex-lethal* (*Sxl*) and the subsequent *transformer* (*tra*) and transcription factors *doublesex* (*dsx*) and *fruitless* (*fru*), which regulates sex determination and differentiation (Cline & Meyer, 1996; Prakash & Monteiro, 2016). Recently, *tra* has been shown to contribute to the size difference between sexes non-autonomously as well as cell-autonomously independent of *dsx* and *fru* (Rideout *et al*, 2015). *Sxl* in specific neurons has also been shown to control female larval body growth but not imaginal tissues (Sawala & Gould, 2017). Thus, while the mechanism of the sexual size dimorphism is gradually revealed, it is not fully known what genes are downstream of the sex genes and whether the sexual size dimorphism and size variation within each sex are regulated by the same or distinct molecular mechanisms.

Recent technological advances have enabled us to investigate the contribution of genomic variation to phenotypic diversity. Genome-wide association studies (GWAS) have identified more than 3,000 genetic variants in more than 700 loci associated with human height (Guo *et al*, 2018). These loci are enriched close to genes implicated in skeletal growth, cartilage proliferation and differentiation, and Wnt, mTOR and epidermal growth factor receptor (EGFR) signaling pathways (Lango Allen *et al*, 2010; Wood *et al*, 2014; Marouli *et al*, 2017). Twin studies have shown that human height has 80% of heritability (Silventoinen *et al*, 2003; Perola *et al*, 2007). However,

1 Institute of Molecular Systems Biology, ETH Zurich, Zürich, Switzerland
2 Laboratory of Systems Biology and Genetics, Institute of Bioengineering, School of Life Sciences, Ecole Polytechnique Fédérale de Lausanne (EPFL) and Swiss Institute of Bioinformatics, Lausanne, Switzerland
3 Faculty of Science, University of Zurich, Zurich, Switzerland
*Corresponding author. Tel: +41 44 633 39 45; E-mail: okada@imsb.biol.ethz.ch
**Corresponding author. Tel: +41 44 633 36 88; E-mail: hafen@imsb.biol.ethz.ch

in the GWAS meta-analyses on human height, ~ 9,500 single nucleotide polymorphisms (SNPs; at $P$-value $< 5 \times 10^{-3}$) only explain ~ 29% of the phenotypic variance (Wood *et al*, 2014). The variation in height is thought to be substantially controlled by environmental factors such as economic (e.g., grain prices), social (e.g., psychosocial stress), and life-historical conditions (Stulp & Barrett, 2016). Advantages of GWAS in model organisms are that it is easy to control environmental influences and to validate the results by performing genetic manipulations. Using the *Drosophila* genetic reference panel (DGRP) that consists of fully sequenced, well-characterized inbred lines derived from a natural population (Mackay *et al*, 2012; Huang *et al*, 2014), we have previously characterized the variation of wing and body size traits using our carefully designed culturing protocol. This allowed us to control for known covariates of size such as temperature, humidity, day–night cycle, and crowding, thus maximizing expression of the genetic effect on size variation. Using this protocol, we performed GWAS to identify size-associated loci against several selected size traits (Vonesch *et al*, 2016). The genetic variants identified were predominantly located in intergenic regions close to genes that do not belong to conserved growth pathways. Subsequently, to identify functional molecular determinants of size variation, we performed proteome-wide association studies (PWAS) in 30 DGRP lines that showed a maximal difference in wing size (Okada *et al*, 2016). We measured protein abundance in wing imaginal discs, the larval wing precursor tissue, with high precision using the SWATH (sequential, windowed acquisition of all theoretical masses) mass spectrometry (Gillet *et al*, 2012; Röst *et al*, 2014; Wu *et al*, 2014; Guo *et al*, 2015; Liu *et al*, 2015; Williams *et al*, 2016; Ludwig *et al*, 2018). We showed that size-associated proteins form tight co-variation clusters that are enriched in fundamental biochemical processes including cell cycle, protein metabolism (translation, folding, and degradation), and glucose metabolism (glycolysis and oxidative phosphorylation). Here, we add a transcriptome layer to the three-layer omics (genomics, proteomics, and phenomics) data on wing growth. By performing multiple association studies among the layers, we now have a more comprehensive view on the genetic architecture of size control (Fig 1A).

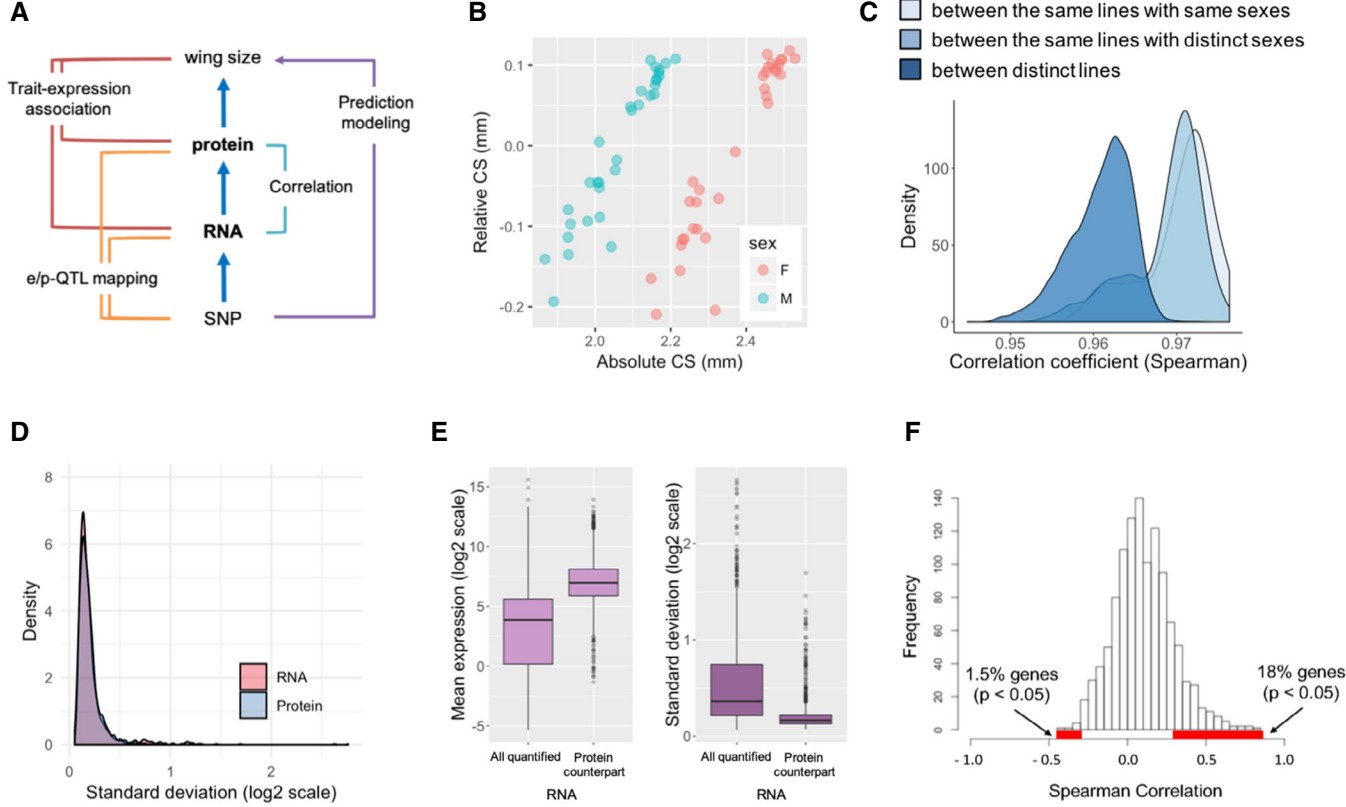

**Figure 1. Variability of wing disc RNA levels caused by a natural genetic diversity.**

A  Analytical scheme of multi-layer systems omics. Multiple association studies among four layers between genotype and wing size phenotypes are performed and, based on the information, a prediction model is constructed.

B  Wing size variation in absolute centroid size (CS) and relative CS (adjusted for body size).

C  Reproducibility of RNA-seq experiment. Pairwise Spearman correlation coefficients between RNA levels showed higher correlations among biological replicates than among non-replicates.

D  Comparable variability of RNA and protein levels for genes quantified at both levels.

E  RNAs of the genes quantified at both RNA and protein levels have the largest means and smallest variabilities among the whole RNAs quantified. The horizontal lines in the boxplot indicate 25th, 50th, and 75th quartiles.

F  Weak correlation between RNA and protein levels. The median Spearman correlation coefficient was 0.1 and only 18% of genes compared showed a positive correlation with statistical significance ($P < 0.05$).

# Results

## Quantification and characterization of wing disc transcriptomes

To perform transcriptome-wide association studies (TWAS), we selected 32 extreme lines (16 big and 16 small wing DGRP inbred lines, mostly overlapping with lines used in PWAS) to maximize the wing size variation (Fig 1B). Wing size was defined as in our previous studies by centroid size (CS), a standard size measure in geometric morphometrics (Okada *et al*, 2016; Vonesch *et al*, 2016) (Dataset EV1). Absolute CS (absCS) is proportional to wing area and reflects the sexual dimorphism of wing size (Fig 1B). Relative CS (relCS) is adjusted for body size using the interocular distance (IOD) as a reference. Flies were grown in the same environmental condition (temperature, humidity, food, etc.) as in PWAS. We dissected third instar larvae and collected wing imaginal discs separately for each line and sex. Biological duplicates were prepared for each case, amounting to 128 wing disc samples in total on which RNA sequencing was performed. Subsequent data processing (see Materials and Methods) identified and quantified 10,017 RNAs from the single organ (Dataset EV2). The reproducibility of the data was evaluated by comparing the correlation of RNA levels between biological replicates and non-replicates (Fig 1C). The Spearman correlation coefficients between biological replicates (individuals with the same genomes) were high (median = 0.971), confirming that environmental influences in the experiment have been successfully minimized. The correlation between distinct sexes (individuals that differ in sex chromosomes only) was slightly lower, and the correlation between non-replicates (individuals with distinct genomes) showed a shifted, distinct distribution, indicating detectable, genetically caused variation in RNA abundance.

It has long been assumed that RNA levels are proxies for protein levels. Our omics dataset is suitable to test this. We first compared the total variability of RNA and protein expression. Interestingly, the overall standard deviation (SD) distributions for genes whose expression was quantified at both levels (1,213 genes × 56 sex/line combinations) did not show a difference (Fig 1D). We found that these RNAs have the highest mean expression and the smallest SD among all the RNAs quantified (Fig 1E), indicating that these genes are abundantly expressed and the levels are relatively invariable among individuals in the population. We next examined the distribution of the gene-based correlation between the levels of RNAs and proteins by comparing both levels across the 56 biological conditions for each gene. The median Spearman correlation coefficient was 0.1 and only 18% of the genes showed a positive correlation with statistical significance (Spearman *P*-value < 0.05), indicating a relatively weak correlation, which is consistent with recent studies (Ghazalpour *et al*, 2011; Schwanhäusser *et al*, 2011; Skelly *et al*, 2013; Wu *et al*, 2014; Liu *et al*, 2016; Williams *et al*, 2016) (Fig 1F).

## Biological processes associated with wing size traits

To identify size trait-associated transcripts, we performed association studies between RNA abundance and wing size traits (Dataset EV3). First of all, to identify transcripts associated with the entire wing size variation encompassing both sexes (Fig 2A, association type 1), RNA level was regressed on absolute wing size. Subsequent multiple testing correction using the Benjamini–Hochberg method (Benjamini & Hochberg, 1995) identified 679 and 1092 RNAs at a false discovery rate (FDR) of 10 and 20%, respectively (Fig 2B). Next, to identify transcripts associated with size variation within sexes (Fig 2A, association type 2), RNA levels were regressed on absolute or relative wing size adjusting to sex. After multiple testing adjustment, 113 and 40 RNAs were found to be associated with between-line absolute/relative wing size variation, respectively, at FDR < 20% (Fig 2C). We also performed the same association analyses between protein abundance and wing size traits (Fig EV1A–C and Dataset EV4). We compared RNA and protein levels among genes that were co-identified at both levels and found that less than a half of the size-associated RNAs continue to be associated at their protein levels and many of the size-associated genes are protein level-specific (Fig EV1D), which is consistent with the finding of the relatively weak correlation between RNA and protein levels (Fig 1F). The size-associated RNAs (and proteins) identified in the two ways mentioned above showed some overlaps with each other but the majority of them was exclusively assigned to the first type of association at both expression levels (Figs 2A and EV1A). It is conceivable that these type 1-exclusive RNAs (and proteins) were expressed at two differential levels corresponding to each sex, but invariant within each sex because they were not selected for association type 2 (Figs 2A and EV1A, association type 3). The analysis of variance (RNA/protein level regressed on sex) indeed confirmed significantly distinct expression levels between sexes for most of the RNAs (and proteins) with type 1-exclusive association (Appendix Fig S1). These results suggest that a large part of the size-associated genes is involved in sexual size dimorphism.

Aiming at identifying biological processes associated with wing size variation, we first performed gene ontology (GO) enrichment analysis on the whole associated RNAs at FDR < 20% (Fig 2D). The first-ranked process associated with wing size was transcriptional regulation, indicating a considerable involvement of transcriptional variation in size variability. The second-ranked process associated with wing size was imaginal disc-derived wing morphogenesis. We therefore investigated association of previously established, growth-regulatory genes with wing size. To cover all the critical growth-regulatory genes, we used the AmiGO 2 gene ontology database and selected more than 200 genes categorized in either "imaginal disc-derived wing morphogenesis", "imaginal disc-derived wing growth", "imaginal disc-derived wing size", "regulation of growth", "regulation of multicellular organism growth", or "growth factor activity" (Dataset EV5). Surprisingly, many of the canonical growth regulators exhibited type-3 association rather than type-2 association with wing size, suggesting their sexual dimorphic expression but no variation between lines of different sizes (Fig 2E). Plots of the RNA levels against wing size indeed confirmed that their expression levels are significantly different between sexes but invariant between small and large wing lines within each sex. The genes with type-3 association are critical components from various canonical growth pathways including many transcription factors (TFs; Fig 2F); for instance, sd (Scalloped) and Stat92E (signal transducer and activator of transcription protein at 92E) are TFs that determine the outcome of Hippo and JAK/STAT signaling pathways, respectively. dm, also named as Myc, is another TF, homologous to vertebrate Myc proto-oncogene, which is important in cell growth and proliferation. Gap1 is a GTPase-activating protein for oncogenic Ras small

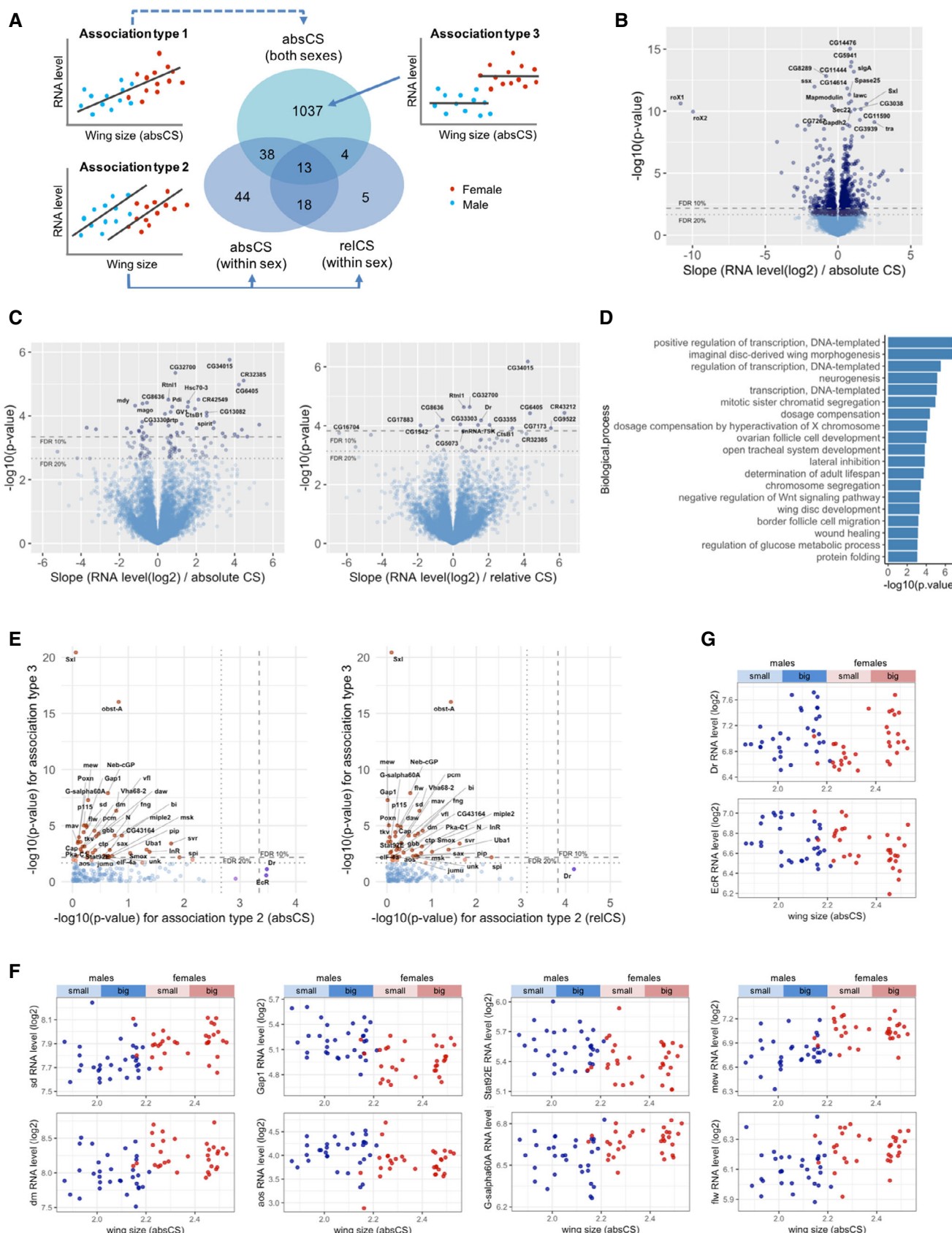

Figure 2.

**Figure 2. Canonical growth regulators are associated with sexual size dimorphism.**

A  Classification of the wing size-associated RNAs at FDR < 20% based on association types. RNAs that exclusively belong to the association type 1 (without overlaps with association type 2) have a type-3 association with wing size.

B  Association of RNAs with the whole wing size variation that encompasses both sexes. Volcano plot of *P*-values against the slope of the fitted lines. The horizontal lines indicate 10 and 20% FDR thresholds.

C  Association of RNAs with between-line wing size variation (absolute/relative CS, adjusted for sex). The horizontal lines indicate 10 and 20% FDR thresholds.

D  GO enrichment analyses on the wing size-associated RNAs (at FDR < 20%). Biological processes associated with wing size at FDR < 0.1 are shown.

E  Association of canonical growth regulator transcripts with wing size (absCS and relCS). *P*-values from the tests for association type 2 (ANCOVA) and type 3 (ANOVA on sex) are plotted. The dashed and dotted lines indicate 10 and 20% FDR thresholds, respectively. Note that canonical growth genes mostly exhibit type-3 association.

F  Sexually dimorphic expression pattern of canonical growth genes. Plots of RNA levels confirm type-3 association with wing size.

G  Exceptional growth genes that are associated with between-line size variation (association type 2).

GTPase, which is essential in terminating Ras signaling. aos (argos) is an antagonist of EGFR signaling that binds to the receptor and reduces the EGF action. G-salpha60A (G protein alpha s subunit) stimulates adenylate cyclase, the cyclic adenosine monophosphate (cAMP)-generating enzyme. Moreover, many pathway components from the TGFβ or decapentaplegic (DPP) signaling also displayed a type-3 association, which includes morphogens (mav/maverick, gbb/glass bottom boat, daw/dawdle), receptors (tkv/thickveins, sax/saxophone), SMAD (Smox/Smad2), and a target (bi/optomotor-blind; Appendix Fig S2). In contrast, only two genes showed type 2-association (Fig 2E). Dr (Drop) is a homeodomain protein known to regulate dorsal–ventral patterning (Milán *et al*, 2001), which has not been implicated in size regulation. Dr showed a strong positive association with between-line size variation (Fig 2G). EcR (ecdysone receptor) is a nuclear receptor of steroid hormone ecdysone, known to regulate developmental transitions such as larval molting and metamorphosis. EcR is an exceptional, canonical growth regulator that showed a type-2 association. We found that the genes with type-3 association are spread around the genome, indicating that the sexually dimorphic expression is not due to direct effects related to dosage compensation on the X chromosome (Appendix Fig S3). Especially, Wnt signaling pathway was explicitly listed in the GO analysis (Fig 2D). The GO title includes "negative regulation" seemingly because the associated genes contained many negative regulators of Wnt signaling such as Coop (corepressor of Pangolin), Roc1a (regulator of cullins 1a), drl (derailed), nkd (naked cuticle), CG17278, and Stat92E. However, their expression levels were found to correlate negatively with wing size (Fig EV2A), which may suggest an overall positive association of Wnt signaling with wing size. We again found that many of the Wnt signaling components curated in the AmiGO 2 gene ontology database also show a type-3 association (Fig EV2B). Plots of their RNA levels confirmed the intra-sexual invariance and inter-sex difference in the expression (Fig EV2C). Roc1a and Rho1 only exhibited a weak type 2-association (Fig EV2D).

To examine size-associated biological processes at the protein level, GO enrichment analysis was performed on the whole associated proteins at FDR < 20%. The top-ranked processes were various cellular processes including protein metabolism, cell–cell junctions, cytoskeletal structures, subcellular trafficking, and RNA splicing (Fig EV1E). We then investigated association of canonical growth regulators with wing size at the protein level. Among the growth-regulatory gene list mentioned above, 26 proteins were identified and quantified in PWAS. This smaller number of identified proteins illustrates that many of the canonical growth regulators are presumably expressed at too low expression ranges to be detected by mass spectrometry whose detection is biased toward abundant proteins. Consistent with the RNA-level results, all proteins associated with wing size exhibited a type-3 association (Fig EV3A), which includes Wnt signaling components: wls (wntless: regulator of Wnt protein secretion) and osa (antagonist of Wnt target gene expression). Interestingly, plots of their RNA and protein levels revealed that their sexual dimorphic expressions are specific to protein levels while Vha68-2 (V1 subunit of vacuolar ATPase) and ctp (cut up: subunit of dynein and myosin V) showed sexual dimorphism at both levels (Fig EV3B). Only a single protein (dp: DP transcription factor) displayed a type-2 association (Fig EV3C).

Overall, these findings are important because extensive genetic studies have established these genes as wing growth-regulatory genes using harsh gene manipulations such as gene knockout/knockdown but we did not have evidence that they contribute to the variation of wing size in natural conditions where the perturbation in gene expression is relatively small. Our data suggest that these canonical growth-regulatory genes and the corresponding pathways collectively participate in the generation of the size variation in natural settings. Furthermore, to our surprise, the results also suggest that they mainly regulate the sexual dimorphism of wing size, but not between-line wing size variation.

Other interesting processes associated with wing size enriched at both RNA and protein levels are mitosis-related processes (Figs 2D and EV1E) because distinct mitotic cell cycle regulation may cause changes in the size and number of cells in the wing. We investigated the association of cell size and cell number in the wing with wing size, based on wing morphometrical data from the lines with extreme small and large wing sizes, which were obtained in our previous study (Okada *et al*, 2016). The cell size in females was significantly larger than in males (Alpatov, 1930), while cell size within each sex did not differ significantly between the smallest 5 and largest 4 wing lines (Appendix Fig S4A). The analysis of covariance clearly illustrated that wing area is proportional to the total cell number in the wing for each sex ($R^2$ = 0.90; Appendix Fig S4B). It indicates that the sexual size dimorphism occurs due to larger size and number of cells in females, while between-line size variation is generated mainly via the cell number variation in the wing. Glucose metabolism was also enriched at both RNA and protein levels (Figs 2D and EV1E). Our previous PWAS showed that, while protein expression in the consecutive subprocesses of glucose metabolism (glycolysis, pyruvate dehydrogenase complex, and TCA cycle) is positively correlated with wing size (Appendix Fig S5A–C), protein levels in the subsequent respiration process, however, exhibit a negative correlation to size (Appendix Fig S5D), indicating an increased use of glycolysis and a reduced use of respiration in the

larger tissues. This finding suggests a similarity to the well-known Warburg effect observed in many cases of highly proliferative cancer cells (Warburg, 1956). Surprisingly, our TWAS now revealed that RNA expression in all the four subprocesses above has a positive correlation with wing size (Appendix Fig S5A–D), highlighting an inverse expression trend between RNA and protein levels in the respiration process only. This transition should occur in a post-transcriptional manner. This observation may provide us with a hint on the mechanism of the long-puzzling Warburg effect (Heiden et al, 2009).

### Functional validation of novel growth-regulatory genes

Many of the wing size-associated RNAs and proteins, especially ones that exhibited a type-2 association with wing size, have not been implicated in the regulation of growth/size. GO analyses of the genes with a type-2 association suggest their enrichment in various cellular functions such as subcellular trafficking, cytoskeletal organization, protein metabolism, immune response, and glycolytic process (Appendix Fig S6). We selected 20 size-associated genes, mostly top-ranked genes with a type-2 association at both/either of expression levels (Fig 2C and EV1C), for which RNAi lines are available, and tested their size-regulatory function (Appendix Fig S7). To validate the function of the genes in wing size, we performed transgenic RNAi using a posterior (P) compartment-specific Gal4/UAS system. The target genes were knocked down by their specific UAS-RNAi transgenes in the P compartment of the wing under the control of hedgehog Gal4 (hh-Gal4; Fig 3A). This system is superior to other systems in that RNAi occurs in the P compartment alone and the untreated, anterior (A) compartment (the other half of the wing) serves as an internal control for size measurement. The boundary between the two compartments is known to prevent intermixing of the cells from the two compartments. The size comparison between the two compartments in the same wing minimizes the influence of environmental confounders that usually occur among different individuals, and also allows an intuitive recognition of a size change. Furthermore, the A compartment provides structural support for the wing and suppresses wing deformation, which provides a useful framework for evaluating the ability of a gene to induce a size change. We first performed RNAi using a single RNAi line per gene. Wing deformation occurred only for a single gene (Fig 3B). RNAi of 4 genes induced lethality, implying their vital function in addition to their possible size regulation (Fig 3B). RNAi of 14 genes allowed precise measurements of size and 13 genes showed significant size changes. To check the reproducibility of the experiment, we then repeated RNAi of the 14 genes using an independent RNAi line or two more lines (if available) for each gene. The combined results include 3 RNAi cases for 9 genes and 2 cases for 5 genes (Fig 3C and Dataset EV6). All the 4 (negative) control cases resulted in the same range of P/A compartment size ratio, indicating a high reproducibility of the experiments. All 3 cases of CG14207 knockdowns resulted in the same range of the P/A size ratio as the controls (even though the P-value of one case claims a significant change), suggesting that CG14207 may not contribute to wing size variation. In contrast, all other RNAi cases on 13 genes exhibited significant size changes. Knockdown of 10 genes (out of the 13) consistently caused reduced P compartment sizes (even though the strength of the effects varies for some

genes), suggesting that these genes are positive growth regulators. However, knockdown of the other three genes (CG7173, Ppn, and yip2) showed significant size changes in opposite directions, depending on lines used. Plots of RNA/protein levels of the 13 genes against wing size variation in the natural population revealed that all the genes except Ppn are positively correlated with wing size, suggesting that only Ppn is a negative growth regulator and the others are positive regulators (Fig EV4). This supports the notion that the 10 genes with consistent size reductions by RNAi function as positive regulators. Among the 3 genes that showed inconsistent RNAi effects, the plots support CG7173 and yip2 to be positive regulators and Ppn to be negative regulators, but more detailed assessments are needed to determine the direction in the size-regulatory effects of the three genes. A close examination of the wing shape clearly demonstrates a size-regulatory function of the genes (Fig 3D). The boundary of the two compartments is close to a straight line in all control cases for KK, GD, and TRiP lines. However, the boundary lines gradually bent more toward the P compartment as the size-reducing effect of the genes becomes larger. This probably occurred due to the reduced growth rate in the P compartment, which bent the A compartment at the normal growth rate over the P compartment. This high rate of true positives (13/14 = 93%) suggests other untested, size-associated genes to be novel growth regulators.

### Genetic association of RNA and protein expressions

We have identified transcripts and proteins associated with the variation of size traits. We now aim to identify genes whose variation in expression is genetically associated. First, with the aim of identifying genetically regulated RNA expression and its causal genomic variants, we performed expression QTL (eQTL) mapping (association studies between variation in RNA levels and SNP variation). We considered 985,510 SNPs that satisfied a minor allele frequency (MAF) $\geq$ 10% among the 32 lines selected for TWAS. We first roughly estimated eQTL density around gene regions with each sex separately by testing all possible combinations between RNAs (10,017 genes) and SNPs (985,510 SNPs). eQTL candidates with nominal P-value < $1 \times 10^{-8}$ were predominantly located at the start and end sites of transcription for each sex, and the two peaks were spread about 100 kb around the center of the peaks (Appendix Fig S8). Therefore, SNPs located within or $\pm$ 100 kb of the gene region were tested for cis-association and the other SNPs outside the region or on a distinct chromosome were tested for trans-association. We performed a permutation-based multiple testing correction, in which we repeated the tests for each sex separately for 10,000 expression permutations. The procedure identified 3,818 SNPs (associated with 533 RNAs) and 3,218 SNPs (associated with 488 RNAs) as cis-eQTLs for female and male, respectively, at an FDR of 20% (Fig 4A and Dataset EV7). No trans-eQTLs were identified presumably due to the high multiple testing burden. We found that about half (49 and 57% for male and female, respectively) of the cis-eQTLs at FDR < 20% were shared between the sexes (Fig 4B). Nearly half of the genes associated with the eQTLs (39 and 47% for male and female, respectively) showed an overlap between sexes. A closer look at the distribution of eQTLs around the gene region confirmed two peaks for both sexes: one peak at the transcription start sites (TSS) and

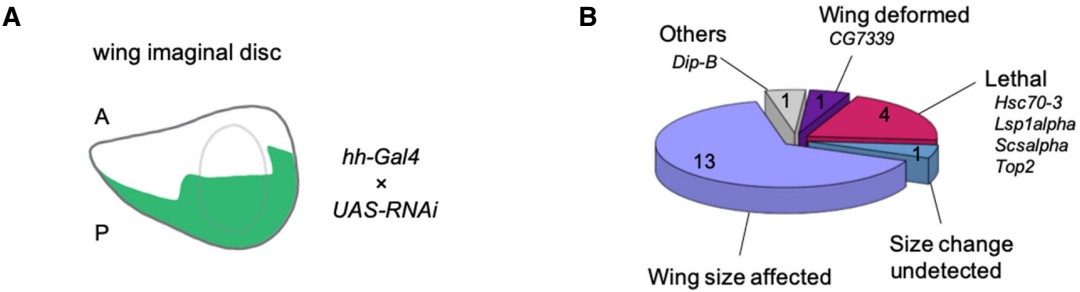

**A**

wing imaginal disc

A

P

*hh-Gal4*
×
*UAS-RNAi*

**B**

Others
*Dip-B*

Wing deformed
*CG7339*

Lethal
*Hsc70-3*
*Lsp1alpha*
*Scsalpha*
*Top2*

Size change
undetected

Wing size affected

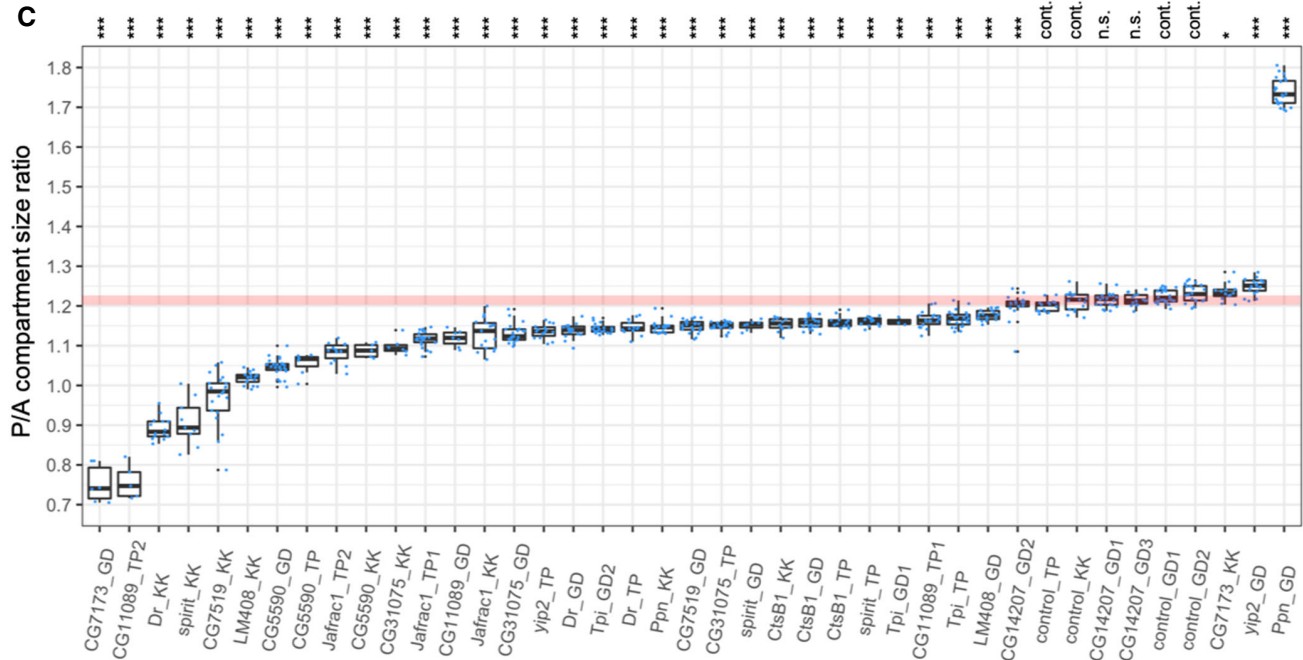

**C**

Genes knocked down and line types used for RNAi

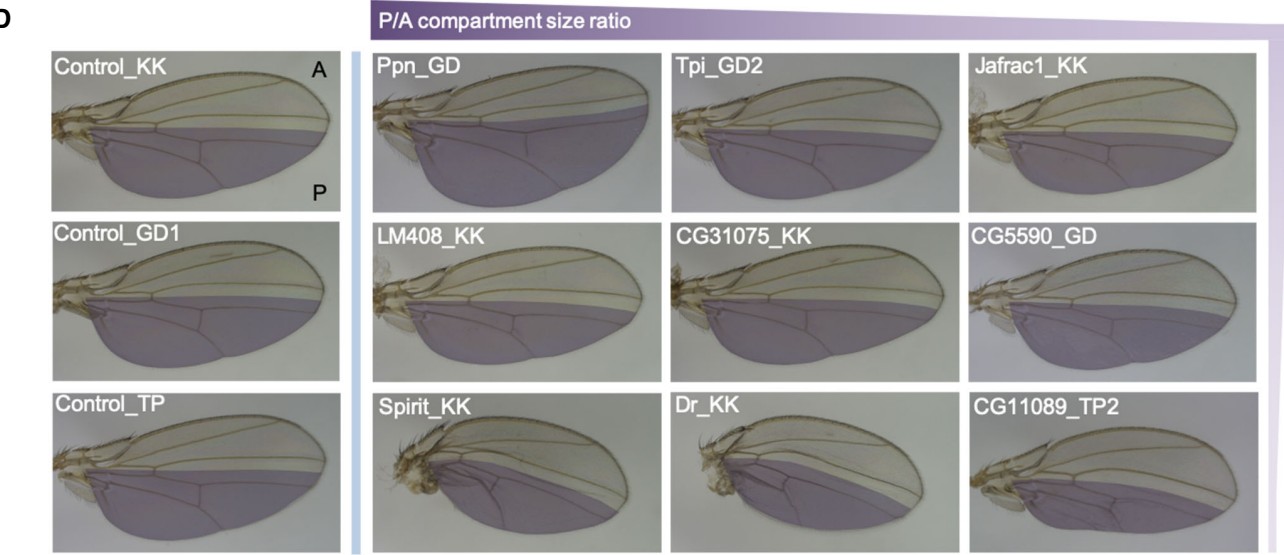

**D**

P/A compartment size ratio

Figure 3.

**Figure 3. Functional validation of novel growth-regulatory genes.**

A   Wing disc compartment-specific RNAi method. Hedgehog (hh)-driven Gal4 knocks down target gene expression only in posterior (P) compartment uniformly. The untreated, anterior (A) compartment functions as an internal control.

B   Classification of observed phenotypes. Structural support by the A compartment allowed a precise measurement of wing size in many (14) genes.

C   The ratio of compartment sizes (P area divided by A area) is plotted. The 14 genes were knocked down using multiple (2 or 3) RNAi lines per gene. The x-axis indicates RNAi case IDs that represent the genes knocked down and the RNAi line type (KK, GD, or TP as TRiP). Detailed information is provided in Dataset EV6. The horizontal line indicates the size ratio in the negative control cases. Wilcoxon rank sum tests evaluated the statistical significance against controls (*$P < 0.05$, ***$P < 0.001$, n.s.: not significant, cont.: control cases). The horizontal lines in the boxplot indicate 25th, 50th, and 75th quartiles.

D   Adult wings with RNAi manipulation in the P compartment (shaded area). Note that the boundary lines between A and P compartments, which are relatively straight in the control cases, bent toward the P compartment due to the reduced growth and size in the P compartment.

the other peak at the transcription end sites (TES), suggesting a modified transcription efficiency and/or RNA metabolism by the eQTLs (Fig 4C and Appendix Fig S9).

Next, we performed protein QTL (pQTL) mapping (association studies between variation in protein levels and SNP variation), in which we tested the whole proteome (1,476 proteins) unlike our previous pQTL mapping where the size-associated proteins only were tested (Okada *et al*, 2016). We here considered 879,102 SNPs that satisfied a minor allele frequency (MAF) $\geqq 10\%$ among the 28 lines selected for PWAS. As with eQTL mapping, SNPs located within or $\pm$ 100 kb of the gene region were tested for cis-association. Permutation-based multiple testing adjustment revealed 23 cis-pQTLs (associated with 10 proteins) in females at FDR < 20% and five cis-pQTLs (associated with two proteins) in males at FDR < 30% (Fig 4D and E and Dataset EV7). It is conceivable that the smaller numbers of pQTLs compared to the eQTL numbers stem from the smaller variability of protein levels (Fig 1E) and the smaller sample size in PWAS. Among these pQTLs identified, seven pQTLs (associated with three proteins) and one pQTL (associated with a single protein) are also eQTLs for females and males, respectively. The three genes with multi-layer QTLs are Got2, P5CDh1, and Dhod, which are all mitochondrial proteins that regulate the metabolism of glutamate, proline, and pyrimidine, respectively.

## Genetic control of wing size

We have shown that wing size variation within each sex is associated (via type-2 association) with the expression of novel growth-regulatory genes. To identify growth regulators that mediate genetic information to size phenotype, we searched for "mediator" genes that exhibit triangular associations among three layers (genotype–gene expression–wing phenotype; Fig 5A). We first searched for RNAs and proteins that were co-associated with both genomic and wing size variation. Nine and seven RNAs (but no proteins) were identified as mediator gene candidates for females and males, respectively, at the 20% FDR criteria for both the eQTL mapping and the trait–expression association with association type 2 (Fig 5B). We revealed that, among the mediator candidates, 8 and 5 RNAs for females and males, respectively, are associated with eQTLs that are also associated with either or both of wing size traits (absCS and relCS; Fig 5B). We realized that the previously uncharacterized *CG34015* that encodes the RNA best associated with both wing size traits through a type-2 association (Fig 2C) is linked to eQTLs that exhibit the best associations with wing size traits (Fig 5B). We performed compartment-specific RNAi of CG34015 in the wing by crossing hh-Gal4 driver line with an RNAi line for

CG34015 and confirmed that CG34015 positively regulates wing size in both sexes (Fig 5C and D). Manhattan plots for the genome-wide association of *CG34015* expression showed a strong, exclusive association with a specific genomic region on the X chromosome for both sexes (Fig 5E). Zoomed plots revealed that these eQTLs are mainly located upstream of the gene region of *CG34015*, and reside in the introns of Rbp2 genes (Fig 5F).

We have shown that expression levels of canonical growth regulators do not show type-2 association with wing size indicating that their RNA and protein abundance is unable to predict between-line wing size. Thus, we next investigated whether we could predict wing size from the expression levels (or eQTLs) of the mediator genes identified above. We first examined 13 eQTLs linked to the mediator *CG34015*. These loci exhibited a strong linkage with each other (Fig 5G). The variation in one of the SNPs (X_16344746) well reproduced CG34015 RNA levels and wing size for both sexes (Fig 5H and I). We then asked whether the SNP could predict the population-wide wing size variation. The SNP variation was successfully associated with wing size variation among 143 DGRP lines, for which wing size was measured in our previous study (Vonesch *et al*, 2016), with a predictability of $R^2 = 0.08$ or 0.11 (for female or male; Fig 5J). We then investigated the integration of genetic effects from the multiple mediator-linked eQTLs and its wing size predictability. We followed genetic information flows from each eQTL, through mediator RNA expression, to wing size traits in 32 lines (Fig 6A). We considered different sets of mediator RNAs and the SNPs for each biological condition (male or female, wing size traits: absCS or relCS) since RNAs and SNPs relevant for each condition are distinct as revealed in Fig 5B (Dataset EV8): The genetic perturbation (by the presence of an alternate SNP) affects expression of the corresponding RNA positively/negatively as revealed in the eQTL mapping. Subsequently, the perturbed RNA level affects wing size positively/negatively as revealed in the trait–expression association study. The final effect of each eQTL on wing size of each line is depicted in the bottom panel of Fig 6A. To integrate all the genetic effects, we simply added individual eQTL effects for each line, assuming that each effect has a unit strength with a positive/negative direction. The combined net effect showed a good rank correlation with wing size variation for both sexes and both size traits (Spearman's $\rho = 0.58$–0.72; Fig 6B). We then tested the predictability of the multiple mediator-linked genotypes using an independent set of 111 lines from DGRP. Even though the lines have much lower size variation compared to the 32 lines of the extreme wing sizes (Appendix Fig S10A), the model successfully predicted wing sizes of the medium-ranged 111 lines at the Pearson correlation of 0.14–0.20 with statistical significance (P-values < 0.05) for all conditions (except a case of male and relative wing size,

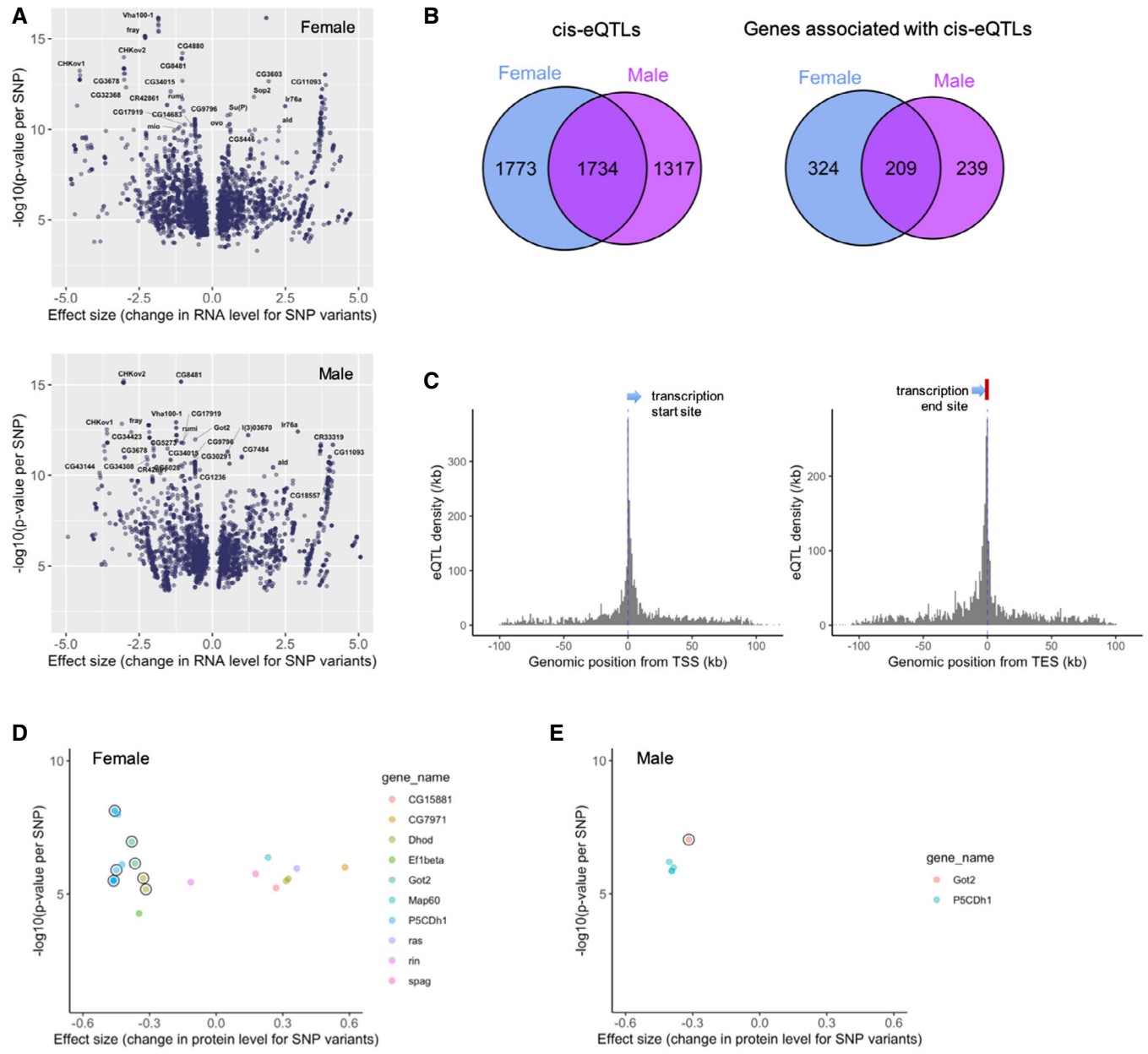

**Figure 4. Genetic association of RNA and protein expressions.**

A   Cis-association of SNP variation with RNA expression for females and males separately. *P*-values for eQTLs at FDR < 20% are plotted against the effect size of the SNPs. The top SNPs are labeled with the corresponding gene names.

B   cis-eQTLs and the associated genes identified in females, males, and in both sexes (FDR < 20%).

C   eQTL density plots. The number of eQTLs per kb is plotted against the genomic location relative to the start and end sites of transcription (TSS and TES) in females.

D, E   Cis-association of SNP variation with protein expression for females and males. *P*-value for each pQTL is plotted against the effect size of the SNP. pQTLs at FDRs of 20 and 30% are depicted for females and males, respectively. The corresponding proteins are designated with distinct colors. The circle indicates that the pQTL is also eQTL for the same gene.

$P = 0.078$; Appendix Fig S10B). Finally, the model predicted the population-wide variation of absCS and relCS from the combined 143 lines for each sex at the Pearson correlation of 0.29–0.36 (Fig 6C). While the causal relationship remains to be determined, the result indicates that the mediator-linked genotypes have a certain level of predictability for between-line wing size variation.

## Discussion

It has long been assumed without evidence that canonical growth regulators regulate size variation in natural populations. Our study has shown that expressions of canonical growth regulators including critical components of conserved growth-regulatory pathways

such as Myc, Ras, Hippo, Wnt, EGFR, JAK/STAT, TGFβ, and GPCR, and integrin signaling pathways are indeed associated with size variation. However, to our surprise, our data indicate that canonical growth regulators primarily regulate sexual size dimorphism, rather than between-line size variation. It appears that RNA/protein level differences between sexes agree with the direction of sexual size

difference (female wing > male wing) for canonical growth genes whose effect on size is evidently known: For instance, positive growth regulators such as dm have higher expression levels in females that have larger wings, and negative growth regulators such as Gap1 and aos have lower levels in females. However, some growth genes seem to show an opposite behavior from expectation:

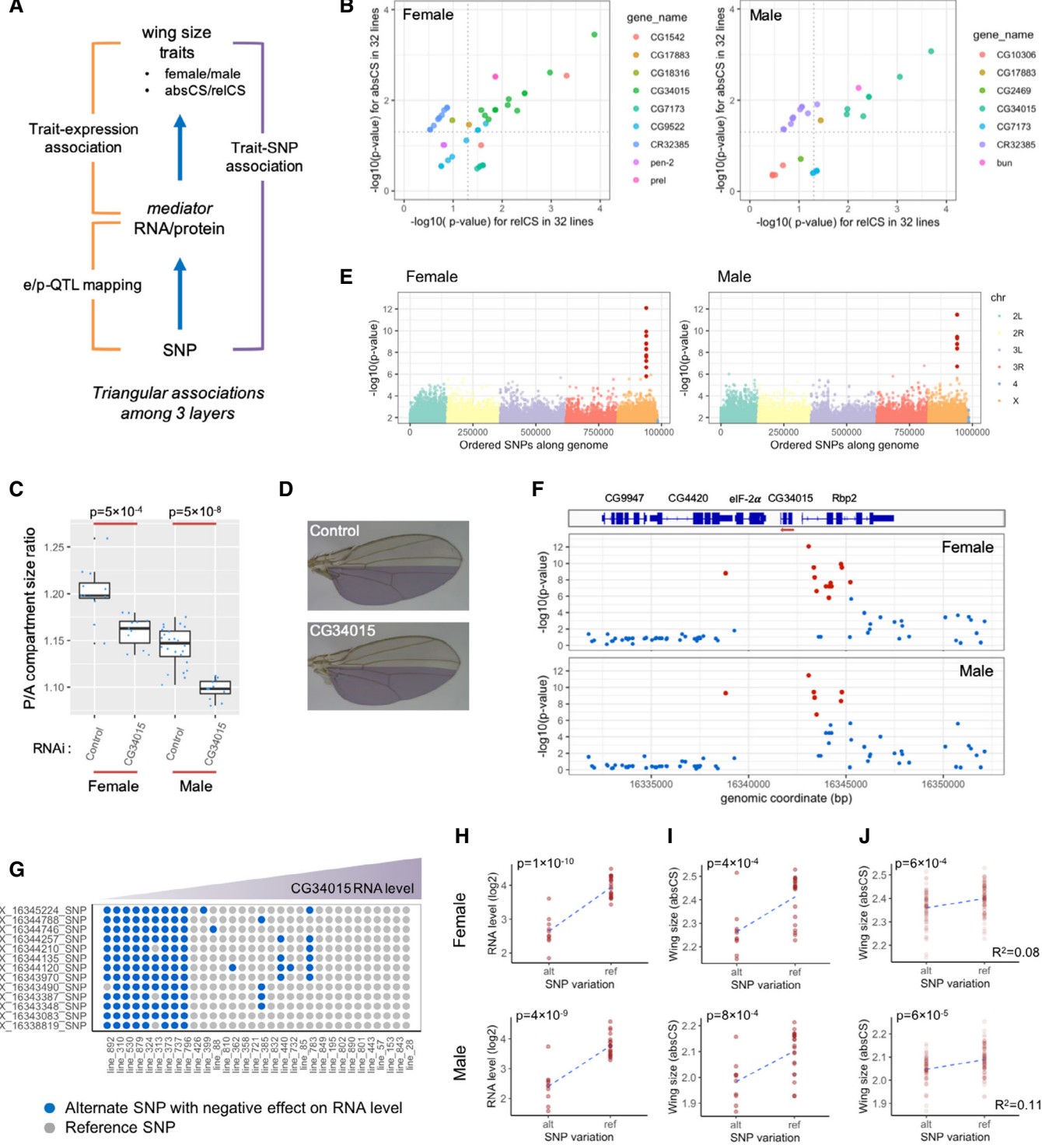

**Figure 5.**

**Figure 5. Genetic information flow through CG34015 to wing size phenotype.**

A   Mediator genes that possibly transmit genomic information to size phenotype are chosen by selecting genes whose RNA or protein level exhibits a triangular association among genotype, gene expression and wing size traits.

B   Selection of mediator RNAs and the linked eQTLs. RNAs co-associated with SNPs and wing size traits are further tested whether the SNPs are also associated with wing size traits (absCS and relCS) for each sex. The dotted lines indicate a significance threshold ($P = 0.05$).

C, D   RNAi of CG34015 in the wing P compartment significantly reduces the size in both sexes. The images are taken from female wings. Statistical significance was evaluated by Wilcoxon test. The horizontal lines in the boxplot indicate 25[th], 50[th], and 75[th] quartiles.

E   Genome-wide association of CG34015 expression. Manhattan plot shows a strong association of CG34015 RNA with X chromosome in both sexes. *P*-values are plotted against ordered SNPs along the genome. eQTLs associated with CG34015 are designated as red dots.

F   Zoomed plot around the CG34015 gene region on X chromosome. eQTLs linked to CG34015 are mostly located upstream of the CG34015 gene region. The red arrow indicates the transcriptional direction.

G   SNP variation of CG34015-linked eQTLs. SNP states at each eQTL are depicted for each line using different colors (gray dots: reference SNP, blue dots: alternate SNP with a negative effect on RNA level). Lines are ordered from left to right as CG34015 RNA levels increase. Note that the eQTLs are highly linked with each other.

H, I   Segregation of CG34015 RNA levels and wing size (absCS) by the variation of an eQTL (X_16344746_SNP). *P*-values are shown.

J   Association of the eQTL (X_16344746_SNP) with population-wide, wing size variation among 143 DGRP lines. *P*-value and $R^2$ are shown for each sex.

For instance, pro-growth Stat92E (Ekas *et al*, 2010) has lower levels in females. These facts suggest that the final sexual size difference is determined by balancing the outcomes of system-wide interactions between positive and negative regulations controlled by many growth genes and pathways.

It is conceivable that sex-determination genes (*Sxl*, *tra,* and *dsx*) are upstream of canonical growth regulators and regulate sexually dimorphic gene expression. In *Drosophila*, it has been shown that loss of *tra* in female larvae decreases body size and overexpression of *tra* in males increases body size; however, *dsx*, downstream of *tra*, has no effect on body size (Rideout *et al*, 2015). This suggests the occurrence of branched pathways that specialize in size regulation. How the sex-determination pathway interacts with the expression of canonical growth regulators and, more specifically, which sex-determination pathway components (if any) are direct master regulators of the sexually dimorphic expression of growth regulators are of great interest. We found that between-line size variation is mostly associated with novel genes that have never been implicated in growth/size. Performing tissue compartment-specific RNAi of a subset of these new regulators allowed us to validate their size-regulatory function. These findings allow us to construct a new working model for wing size variation (Fig EV5): The size difference between sexes is mainly generated by differential expression/activity of previously known, canonical growth regulators, and size variation within each sex is largely produced by novel growth regulators involved in a wide range of cellular functions.

We note that EcR, one of the canonical growth regulators, is unique in exhibiting a significant association with between-line wing size at FDR < 20%. A molting hormone, ecdysone, and its nuclear receptor EcR have been thought to provide a substantial contribution to *Drosophila* body and organ size (Mirth & Shingleton, 2012). EcR forms a heterodimer with Ultraspiracle (usp) and, without ecdysone, binds to the promotor of target genes and represses transcription (Schubiger, 2005). Once EcR binds to ecdysone, the transcriptional repression by the unliganded EcR-usp complex is eliminated and EcR, recruiting transcriptional activators and co-factors, activates transcription, thus allowing developmental transitions to proceed. Knockdown of EcR has been shown to derepress transcription, which also allows developmental programs to proceed, mimicking ecdysone action (Mirth *et al*, 2009). We now observed the negative correlation of EcR levels with between-line wing size variation. A possible explanation would be that higher levels of EcR require a smaller amount of ecdysone for

developmental transitions through transcriptional activation and, therefore, higher levels of EcR expedite the transitions, resulting in a shorter growth period and smaller tissues. It is also possible that the variability in ecdysone titers secreted from prothoracic glands and its temporal profiles affect between-line wing size non-autonomously. The relative contribution of EcR compared to that of novel regulators that were identified in the study remains to be determined.

In addition to the hormonal systems, insulin-like signaling and mTOR (insulin/TOR) pathway have been thought as the key players in the environmental and physiological control of body/organ size in *Drosophila* (Mirth & Shingleton, 2012). Our list of canonical growth genes includes several key components from the insulin/TOR pathway such as Ilp2 (insulin-like peptide), InR (insulin-like receptor), Pi3K92E (catalytic subunit of phosphatidylinositol 3-kinase), Pten (phosphatase and tensin homolog), Akt1 (protein kinase B), and foxo (forkhead box, subgroup O transcription factor). Among these, InR was identified to be associated with wing size at FDR < 20% (Fig 2E). The positive association with wing size was through type-3 association, suggesting its contribution to sexual size dimorphism. GWAS and eQTL mapping in our studies did not find any SNPs associated in the InR region. However, GWAS of domestic dogs that experienced intensive selection for size have found that alleles at insulin-like growth factor 1 (IGF1) and its receptor (IGF1R: mammalian ortholog of InR) are critical to the size variation (Sutter *et al*, 2007; Hoopes *et al*, 2012). This may indicate that very rare alleles that rarely survive in natural settings were artificially selected in the domestic dog breeding.

We have shown that 9 RNAs exhibit a triangular association between genotype–RNA–wing size phenotype. The best associated RNA is CG34015. The function of CG34015 is unknown in *Drosophila* but a sequence similarity search found Histidine triad nucleotide-binding protein 3 (HINT3) as the mammalian ortholog, which is a highly conserved gene in eukaryotes. Biochemical assays have shown that mammalian HINT3 and HINT1 (28% sequence identity to HINT3) hydrolyze phosphoramidate and acyl-adenylate substrates, but their physiological function is still unclear (Chou *et al*, 2007). It is of an interest that HINT1 has been assigned a variety of roles including apoptotic, transcriptional, and tumor suppressor activities. The gene region of CG34015 is located between two translation initiation factors eIF-2alpha and Rbp2 (also named as eIF4H1) and most of the CG34015-associated eQTLs reside within the gene region of Rbp2, which raises the possibility of genetic

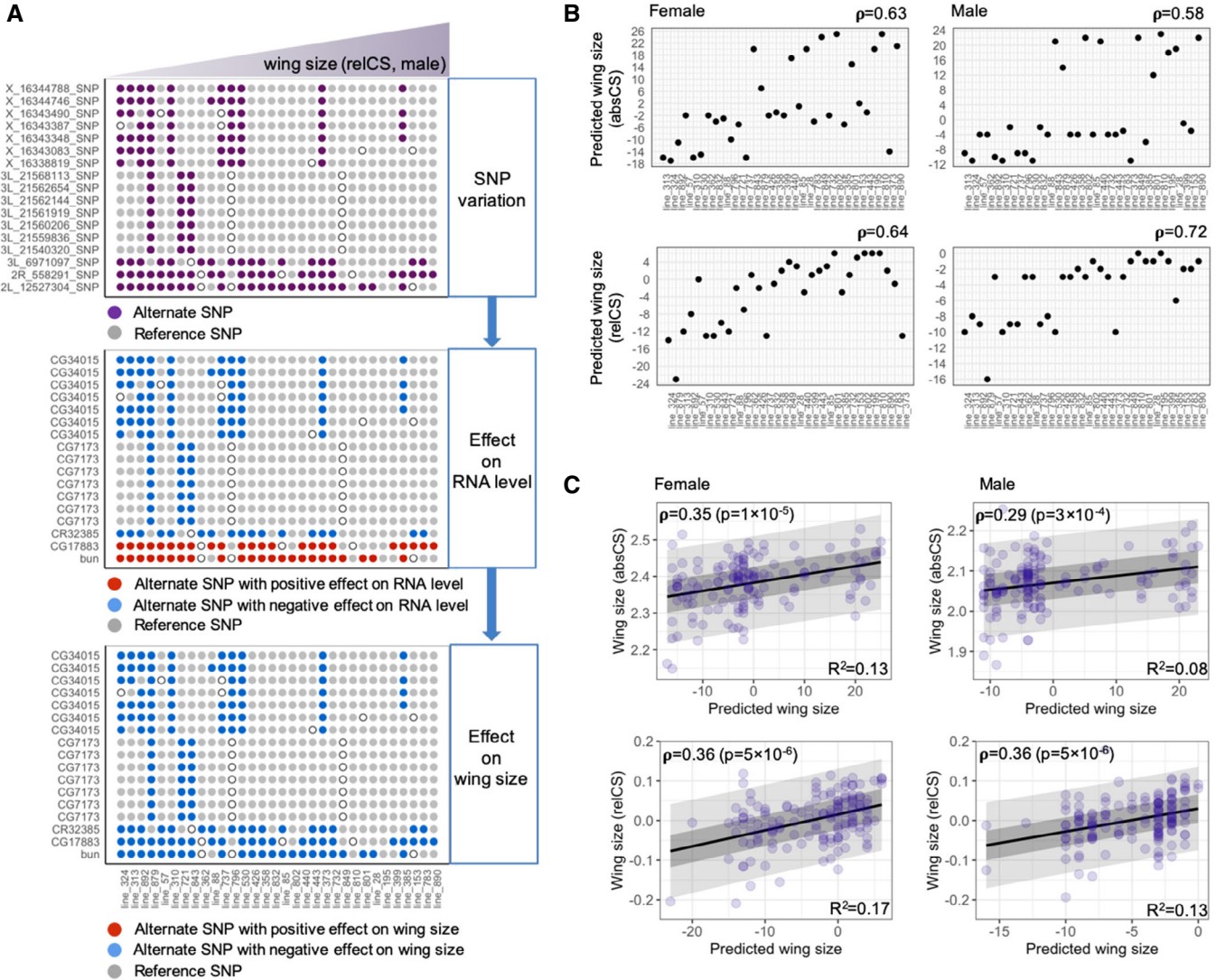

**Figure 6. Wing size predictability of mediator-linked genotypes.**

A  Genetic information flow diagram that models the transmission of genotype information through molecular (RNA) levels to wing size phenotype. The diagram depicts the prediction of relative wing size in male. (Top panel) SNP variation of the eQTLs for mediator genes is depicted for each line. The SNP states are represented by different colors (gray dots: reference SNP, purple dots: alternate SNP, open circle: unassigned SNP status). Lines are ordered as wing size (relCS) in males increases. (Middle panel) Expected effect of each eQTL on the corresponding RNA level is illustrated. The direction of RNA-level perturbation, determined by eQTL mapping, is displayed with different colors (gray dots: no effect by reference SNP, red dots: increased RNA level by alternate SNP, blue dots: decreased RNA level by alternate SNP). (Bottom panel) Predicted effect of each eQTL on wing size perturbation is illustrated. The direction of wing size perturbation, determined by the trait–expression association study, is displayed with different colors (gray dots: no effect by reference SNP, red dots: increased wing size by alternate SNP, blue dots: decreased wing size by alternate SNP).

B  Predicted wing sizes, calculated by simply adding the effects from all mediator-linked eQTLs, are plotted against fly lines ordered as the observed wing size trait (absCS/relCS) increases for each sex. Note good rank correlations between the observed and predicted wing size traits for both sexes (Spearman's $\rho$ = 0.58–0.72).

C  Prediction of population-wide, between-line, wing size variations from the mediator-linked genotypes. Mediator-linked eQTLs relevant for each condition (size traits, sexes) were used to predict wing size traits (absCS/relCS) of 143 DGRP lines for each sex. Spearman correlation coefficient and the *P*-value, and $R^2$ from linear fitting are shown for each case (size traits, sexes). The dark and light gray zones indicate 50 and 95% prediction ranges.

interactions among the three genes. Our previous GWAS and the eQTL mapping in the current study did not find any single SNPs around the Rbp2 region associated with wing size and Rbp2 expression, respectively. However, gene-based GWAS using the sum of chi-square VEGAS method (Liu *et al*, 2010), which evaluates *P*-values considering all SNPs within each gene, identified Rbp2 within the top 20 associations and, therefore, Rbp2 was functionally

tested by RNAi (Vonesch *et al*, 2016). We knocked down the gene level in the whole wing using nubbin-Gal4 driver by crossing with two independent RNAi lines. Rbp2 was among genes with the best reductions in size (~ 10%) in both sexes. The current study revealed that the Rbp2 RNA level exhibited a weak, positive correlation with between-line wing size variation (*P*-value = 0.05 for type-2 association) but the protein level rather showed a sexual dimorphic

expression pattern (*P*-value = 0.78 for type-2 and 0.0055 for type-3 associations;Appendix Fig S11). eIF-2alpha, immediately downstream of CG34015, showed a strong sexual dimorphic expression pattern at both RNA and protein levels (*P*-value = $3 \times 10^{-11}$ and $4 \times 10^{-6}$, respectively, for type 3;Appendix Fig S11). Considering that CG34015 showed the best, positive association only with between-line size differences (*P*-value = $2 \times 10^{-6}$ for type 2, 0.28 for type 3), these results indicate that the two neighboring translational initiation factors are under distinct genetic controls and rather contribute to sexual size dimorphism.

The mutual associations among three layers (genotype–RNA–wing size) for the mediator genes suggest that the genetic variants regulate mediator RNA levels and, in turn, the altered RNA levels control wing size. The causal relationship could be verified by constructing fly lines with an alteration of a single/multiple SNP(s) at the eQTL(s) and measuring the effect on RNA levels and wing size, which remains to be tested but is technically difficult to achieve. We instead constructed a genetic information flow model based on the observed associations and tested whether the mediator eQTL-based genotypes could predict between-line wing size variation at the DGRP population level. We modeled the transmission of individual eQTL effects through RNA to the wing phenotypic level for each line. The simple addition of individual effects had a reasonable prediction power for the population-wide wing size variation for each sex with an R-squared ranging between 0.08 and 0.17. The extent of size variation explained by the model is comparable to that of a single GWAS on human height (10% phenotypic variation explained) (Lango Allen *et al*, 2010), but may not be high enough considering the minimized environmental effects in our study. The most probable cause in our opinion is the small sample size that reduces overall statistical power.

A recent technological advancement in omics field provides us with tools by which we can investigate molecular pathways that bridge genetic variation and phenotypic variation. Proteins are expected to be the final determinants of phenotypes, and therefore, protein abundance is functionally more relevant than RNA abundance. Recent large-scale omics studies including the current study have demonstrated a relatively weak correlation between abundance of proteins and RNAs, which implicates distinct genetic architectures between the two molecular layers along the information paths from genotypes to phenotypes. Based on the central dogma of molecular biology, RNA is situated more closely to genomic information than protein. Therefore, it can be expected that genomic variants associate with RNA expression more closely. Indeed, our study has shown concentrated eQTLs at the TSS and TES of transcripts, suggesting genetic controls of RNA metabolism. In the study, by adding the transcriptomic analyses to our previous proteomics analyses, our data have increased the gene coverage that now includes non-coding genes and lowly expressed genes and thus provide a more comprehensive, detailed view on the genetic information paths from genotypes though two molecular layers to size phenotypes.

In summary, our systems approach has shed light on system-wide gene regulatory mechanisms that differentially regulate sex-dependent and sex-independent size variations. It revealed a heavily biased use of canonical growth mechanisms in sexual size dimorphism. This inversely suggests that evolutionary pressure achieved a strict control of growth mechanisms that attains consistent organ and body sizes within sexes and in return ensures sexual size dimorphism.

# Materials and Methods

### *Drosophila* strains

The DGRP lines used for TWAS, listed in Dataset EV1, were obtained from the Bloomington Drosophila Stock Center. The DGRP lines used for PWAS were previously described (Okada *et al*, 2016). The GD and KK lines for RNAi were obtained from the Vienna Drosophila Resource Center (VDRC), which were to knock down *CG7339* (VDRC_ID: 101401), *CtsB1* (108315, 45345), *Hsc70-3* (101766), *CG7173* (107756, 15134), *Dr* (110625, 7791), *CG31075* (101809, 25676), *CG5590* (109310, 45462), *CG7519* (110613, 21652), *Jafrac1* (109514), *LM408* (108218, 49352), *Lsp1alpha* (101101), *Ppn* (108005, 16523), *Scsalpha* (107164), *spirit* (107936, 5497), *CG14207* (31800, 31802, 44831), *Top2* (30625), *Tpi* (25643, 25644), *yip2* (26562), *CG11089* (31420), *Dip-B* (6296), *CG33920* (103447), and *CG1315* (47097). The TRiP lines for RNAi were obtained from the Bloomington Drosophila Stock Center (BDSC), which were to knock down *Dr* (BDSC_ID: 42891), *spirit* (42882), *CG5590* (66929), *CG31075* (50654), *Jafrac1* (34971, 32498), *CtsB1* (33953), *CG11089* (53332, 58121), *Tpi* (51829), *yip2* (36874), *CG34015* (54031), and *mCherry* (35785). *hh-Gal4* was obtained from Genetic Strains Research Center, NIG, Japan.

### *Drosophila* culture and wing disc dissection

Flies were cultured in a standard condition as previously described (Okada *et al*, 2016). Third instar larvae wandering on the wall of the culture vial were dissected in ice-cold Hank's balanced salt solution under the microscope. Wing discs were collected separately for each sex, snap-frozen in liquid nitrogen, and kept at −80°C until use. This process was repeated at different dates and cultures until the total number of wing discs per line/sex/replicate became more than 30 (45 on average).

### Wing disc RNA sample and library preparation

Wing disc samples in tubes kept at −80°C were thawed and centrifuged at 3,400 *g* for 2 min on table-top centrifuges. The supernatant was removed, and the discs in a tube were lysed by pipetting up and down in 100 µl of TRIzol Reagent (ID: 15596018, Life Technologies). The lysate was transferred into the next tube, and wing discs were mixed and lysed. This was continued until the last tube, to have more than 30 wing discs lysed. TRIzol was added so that the lysate volume was to be 1 ml. To perform phase separation, 0.2 ml of chloroform was added and vigorously mixed for 15 s. Following 3-min incubation, the sample was centrifuged at 12,000 *g* for 10 min at 4°C and the upper aqueous phase was transferred into a new tube. To isolate RNA, the sample was mixed with 0.5 ml of 100% isopropanol for 10 min and centrifuged at 12,000 *g* for 10 min at 4°C. After the removal of sup, 1 ml of 75% ethanol was added slowly and incubated for 15 min at −80°C. Following centrifugation at 7,500 *g* for 5 min at 4°C, the sup was discarded. For thorough cleanup, after being dissolved in 100 µl of RNase-free

water, RNA was additionally cleaned using RNeasy Mini Kit (ID: 74104, Qiagen) following the manufacture's instruction and kept at −80°C until use. Two RNA-seq libraries were prepared from 128 RNA samples using the Bulk RNA Barcoding and sequencing (BRB-seq) protocol (Alpern *et al*, 2019), each of which contained cDNAs made from the 3′-end fragments of transcripts differentially labeled for 64 RNA samples. Briefly, BRB-seq uses the concept of early multiplexing to produce 3′ cDNA libraries that allows multiplexing of up to 96 different RNA samples. The protocol is adapted from SCRB-seq, developed for single cell transcription profiling, with several important modifications involving modified oligo-dT for cDNA barcoding and second-strand synthesis with DNA PolII Nick translation. The sequencing library is then prepared using cDNA tagmented by an in-house i7/i7 compatible Tn5 transposase and further enriched by limited-cycle PCR with Illumina compatible adapters.

### RNA sequencing and data processing

Prepared libraries were sequenced for 75 cycles on the Illumina NextSeq 500 (RRID:SCR_014983) in the Gene Expression Core Facility at École Polytechnique Fédérale de Lausanne (EPFL), Lausanne, Switzerland. The raw reads from BRB-seq experiments carry two barcodes, corresponding to the late and early step multiplexing. The late step multiplexing using Illumina indexes is common to standard protocols and is used to separate the libraries. The early barcode is specific to the BRB-seq protocol and is used to separate the multiplexed samples from the bulk data. The first demultiplexing step was performed by the sequencing facility using bcl2fastq software (RRID:SCR_015058). Then, the data consist of two FASTQ files (R1 and R2). The R2 FASTQ file was aligned to the Drosophila melanogaster genome assembly (BDGP Release 5; http://genome.ucsc.edu) using STAR with default parameters prior to the second demultiplexing step. Then, using the BRB-seqTools suite (available at http://github.com/DeplanckeLab/BRB-seqTools), we performed simultaneously the second demultiplexing, and the count of reads per gene from the R1 FASTQ and the aligned R2 BAM files. Raw count data were filtered using the edgeR (RRID:SCR_012802)/limma (RRID:SCR_010943) package in R (Ritchie *et al*, 2015): six samples were removed due to reduced counts and genes with insufficient counts (if the total count was < 70% of sample size) were also removed, which ended up in the table of 10,017 genes × 122 samples. The count data were then normalized to log-transformed CPM (counts per million) using the voom package in R (Law *et al*, 2014). The batch effect was removed by regressing out the first two principal components. RNA levels for each line (32 lines) and sex were calculated as the mean among replicates.

### Trait–expression association studies

All the association analyses were performed in R statistical environment (version 3.5.2; RRID:SCR_001905, http://www.r-project.org). Wing size phenotypes (absCS and relCS) and protein levels were previously measured and calculated (Okada *et al*, 2016; Vonesch *et al*, 2016). To identify RNAs/proteins associated with wing size (via three association types), we applied linear models to each RNA/protein and evaluated the significance level for each association type. In RNA analyses, to account for non-constant variance in

RNA levels intrinsic to count-based data such as RNA-seq data, weights evaluated by the voom package were used in the linear models. To identify RNAs/proteins associated with the whole wing size variation encompassing both sexes (association type 1), a simple regression was performed: RNA/protein level = absCS + ε. To identify RNAs/proteins associated with wing size traits within sex (association type 2), ANCOVA adjusting to sex was performed: RNA/protein level = sex + absCS/relCS + ε. To identify RNAs/proteins that exhibit a sexually dimorphic expression (association type 3), one-way ANOVA was performed: RNA/protein level = sex + ε. Multiple testing correction was performed using Benjamini–Hochberg method. The FDR was estimated by p.adjust() function in R.

### Gene ontology enrichment analyses

Biological processes enriched for the wing size-associated RNAs at FDR < 20% were identified by DAVID 6.8 (RRID:SCR_001881, https://david.ncifcrf.gov/). Processes enriched at FDR (Benjamini) < 10% were depicted. Gene components involved in biological processes in *Drosophila* were identified using AmiGO 2 gene ontology database (RRID:SCR_002143, http://amigo.geneontology.org/amigo/ landing).

### eQTL/pQTL mapping

Genotypes of the 32 lines used in eQTL mapping and 28 lines in pQTL mapping were obtained from the DGRP Freeze 2 (http://dgrp2.gnets.ncsu.edu). The association tests between RNA/protein levels and SNPs for each sex separately were performed using Matrix eQTL package in R (Shabalin, 2012). We used 985,510 SNPs or 879,102 SNPs for each mapping that satisfied MAF ≧ 10% among the 32 or 28 lines, respectively. SNPs located within ± 100 kb of gene regions were tested for cis-association and the rest of SNPs located outside the regions including ones on other chromosomes were tested for trans-association. Multiple testing correction was performed through permutation as previously described (Stranger *et al*, 2007; Massouras *et al*, 2012; Wu *et al*, 2013). Briefly, we repeated the whole tests for 10,000 permutations of RNA/protein expression. For each permutation, the minimum *P*-value was recorded for each RNA/protein. A corrected *P*-value was calculated as the number of minimum *P*-values from the permutations that were smaller than the original *P*-value divided by the number of permutations. FDR was calculated as the ratio of the number of RNAs/proteins expected to pass the corrected *P*-value threshold by chance over the number of RNAs/proteins that actually passed.

### Wing tissue compartment-specific RNAi and wing image analysis

The original *hh-Gal4* line was crossed to *yw* laboratory wild-type strains. Cleaning the chromosomes by recombination in female, *yw*, *hh-Gal4/TM6b* line was created. The P compartment-specific RNAi was performed at 25°C by crossing virgin females of the *hh-Gal4* line to males of respective UAS-RNAi lines described above. The adult female offspring with correct genotype (e.g., no TM6b balancer) were collected and stored at −20°C until wing size measurement. Right wing of the offspring was detached and mounted in a drop of water on a glass slide. The wing images were captured using a

VHX-1000 digital light microscope (KEYENCE). The A and P compartment areas of each wing image were separately measured using the "polygon selections" and "measure" functions in Fiji (RRID:SCR_002285, https://fiji.sc/). Change in the ratio of wing compartment areas (P compartment area divided by A compartment area) relative to the control was tested using Wilcoxon rank sum test (Wilcoxon.test() function) in R. We used UAS-RNAi lines against *CG33920* and *CG1315* genes as reference lines for KK and GD lines, respectively, as their knockdowns had never showed any noticeable size changes in eyes and wings (Vonesch *et al*, 2016; Nowak *et al*, 2018). The UAS-RNAi line against mCherry was used as a reference line for TRiP lines.

## Data availability

All the raw RNA-seq data are available at the ArrayExpress database (RRID:SCR_002964, http://www.ebi.ac.uk/arrayexpress) under accession number E-MTAB-7662. All the raw MS data are stored at the Center for Computational Mass Spectrometry (RRID:SCR_008161, http://proteomics.ucsd.edu) with MassIVE ID: MSV000079202 and MSV000079208 as previously described (Okada *et al*, 2016).

**Expanded View** for this article is available online.

## Acknowledgements

We thank Julie Russeil for technical support on library preparation. We also thank the GECF (Bastien Mangeat) at the EPFL for sequencing. This work was supported by the SNF consecutive grants 31003A_162557 and 31003A_182532 and by the SystemsX.ch grant 51RT-0_145725 to EH, and by a Chan Zuckerberg Initiative (2018-182612 − 5022), a SystemsX.ch (AgingX) grant 51RTP0_151019, as well as EPFL institutional funding to B.D.

## Author contributions

HO designed the study, performed experimental works and statistical data analyses and modeling, and wrote the paper. RY set up and optimized wing image analysis. VG performed RNA-seq data preprocessing. BD provided critical inputs on data analysis. HO, RY, VG, BD, and EH edited the manuscript. EH and HO conceived the project. EH supervised the project.

## Conflict of interests

The authors declare that they have no conflict of interest.

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
