## [Review Process File · Molecular Systems Biology]

Sex-dependent and -independent regulatory systems of size variation in natural populations

Hirokazu Okada, Ryohei Yagi, Vincent Gardeux, Bart Deplancke and Ernst Hafen.

Review timeline:

Submission date:	20 th May 2019
Editorial Decision:	26 th June 2019
Revision received:	23 rd September 2019
Editorial Decision:	24 th October 2019
Revision received:	29 th October 2019
Accepted:	30 th October 2019

Editor: Hirokazu Okada

Transaction Report:

1st Editorial Decision

26th June 2019

Thank you for submitting your work to Molecular Systems Biology. We have now heard back from the three referees who agreed to evaluate your manuscript. You will see from the comments below that the referees find the manuscript to be of interest. They raise, however, several issues, which should be convincingly addressed in a revision.

Without reiterating all the points raised in the reviews below, some of the more substantial issues are the following:

- The comments from referee #1 regarding the data analysis and statistics should be addressed.
- Further experimental validations suggested by referee #2 need to be included to better support the conclusions of the study.

REFeree REPORTS.

Reviewer #1:

This is a very well written paper that is data dense. The topic fits the scope of the journal very well and would likely appeal to the readership. The paper uses the *Drosophila* genetic reference panel (DGRP) as a reference population to dissect the genetics of wing size using a systems biology approach. The data contain several layers. 1) DNA sequences collected by the DGRP; 2) gene expression of both sexes in 32 extreme lines in imaginal discs. 3) protein expression from a previous study in 30 lines; I believe not all of these lines overlap with the 32 RNA expression lines but it's not very clear from the paper. 4) wing size trait data from these selected lines and other lines, also collected from a previous study. Then the authors performed a series of analyses to correlate these different levels of variation and interpret their data using analysis such as pathway enrichment, etc..

The analyses are sound and comprehensive. They found that canonical growth genes don't contribute much to the natural variation. Instead, they found a bunch of non-canonical genes that seem to regulate wing size variation within each sex. The most interesting and valuable part of the paper is the prediction piece, in which they are able to predict the phenotype of wing size by summing up allelic effects (+ or -) of a handful of SNPs that are associated with the trait and RNA expression. There are several main comments I have regarding the analyses and interpretation, which I think are important.

1) Figure 2E: The y axis of this figure is association P values for type 3. However, my understanding is that your type 3 associations were identified through type 1 analysis. So the P values are not for type 3 but type 1. In addition, I think it would make more sense to make a plot of the same x, and y but for all genes (without selecting canonical growth genes). If all genes have the same pattern, then it's not surprising that a selected subset also shares the same pattern. Based on Figure 2A, the majority of genes are type 3, I suspect this to be true. This is very important and affects some of the interpretations.

2) Figure 4A: the x-axis is SNP effect, but the y axis is minimum p value for a gene. Can you plot the P values of SNPs on the y axis instead? I don't see the necessity to use a gene minimum p value.

3) Figure 5A: I don't understand why this figure allows you to display "mediators". Some graph to explain what your definition of mediators is would be helpful.

4) One of the main conclusions of the paper is that a few trait associated SNPs are able to predict phenotypes with good accuracy. $R^2 = 0.16$ translates to correlation of 0.40, which is pretty good compared to genomic prediction in *Drosophila*. However, it's not clear that when prediction was made on 143 lines whether these lines include the 32 lines used in the present study. The correct analysis is to run this with an independent set of lines. If the 32 lines were among the 143 lines, please remove them and re-evaluate the correlation between the predicted and observed wing sizes. This is very important because including the lines in the test set can significantly bias the prediction accuracy upwards.

Minor comments:

Bottom of page 3: "it is fully unknown" -> "it is not fully known"

Page 4: The contrast between

Page 7, Figure 1E: these RNAs have the smallest sd, do they also have the highest mean expression? I wonder what this means.

Reviewer #2:

Okada et al.

This is an interesting manuscript that attempts to analyse the genetic basis for natural wing size variation between *Drosophila* sexes and between different natural strains. Overall, the authors have done a good job of their analysis and presentation of the findings. I recommend publication.

Minor comments:

1. In the abstract, the authors should mention that the Ecdysone Receptor was identified as being differently expressed between fly lines of different wing size, along with other novel genes not previously implicated in size control. The EcR result is important and should be highlighted rather than be buried. It is interesting that size differences between mammals (including sex differences) are also strongly governed by steroid hormones.

2. Is it also worth mentioning that not a single component of the Insulin-IGF1 pathway was identified in this analysis, despite the fact that IGF1R is crucial in variation between species of dogs (artificially selected)? This might be because IGF1R-variation leads to large dogs with short life spans, while natural variation leads to large mammals with long life spans.

3. Did the authors attempt to RNAi their HINT homolog in the wing? Is there any chance that the variation in HINT might also affect the neighbouring downstream gene eIF2a?

Reviewer #3:

A number of genetic studies in the past have identified genes that have the capability to affect tissue size during development if genetically perturbed. Importantly, however, this does not mean that these genes actually do regulate tissue size during development. They might simply be required for growth, but not be the regulatory node that the biological system uses to control growth physiologically. Hence an important open question is what genes control tissue and animal size in vivo in non-experimentally-perturbed conditions. This interesting study starts addressing this issue by looking at the association of mRNA levels in the developing *Drosophila* wing with final wing size, across fly lines and between males and females. The manuscript is well written. The data are interesting. For the most part, the conclusions are solid and well founded (with the two exceptions mentioned below). The approach presented here seems to have been quite successful, identifying 13 new genes (out of 14 tested) that affect organ size. I think this study will be of interest to a wide range of people interested in development, organ size and growth, patterning, metabolic regulation, and signaling pathways that control cell growth and proliferation.

The following results would need to be strengthened:

1. The size phenotypes presented in Fig 3C would need to be confirmed/repeated with independent RNAi lines. Especially undergrowth phenotypes may reflect unhappy cells, and there is a good chance of them being due to off-target effects of the RNAis.
2. The authors identify CG34015 as an important regulator of wing size, but this is not functionally tested. Does modulation of CG34015 expression (either RNAi or overexpression) affect wing size? Since the nearby SNPs are actually in the neighboring gene *Rbp2*, it would be good to also test this one.

Minor Issues

1. Fig 1F - it is not completely clear from the main text and figure legend how this analysis was done. Were RNA and protein levels correlated within one fly strain, thereby yielding a correlation coefficient, and this was done for all 32 fly strains? (In which case I would expect the total sum of correlation coefficients in the curve in Fig 1F to be 32. However, it looks like the distribution of roughly 1000 correlation coefficients is shown in 1F.) So instead, was the correlation performed comparing RNA and protein for one gene across the 32 fly strains, thereby yielding a correlation coefficient for each gene ?) This should be spelled out a bit more.
2. It is not very clear from the text how the 20 size-associated genes to test were selected. For instance, *yip2* appears at position 392 on the Suppl. Table 3 list when sorted by FDR. It would be good to explain in more detail how these 20 genes were selected, because by showing they were not cherry-picked would strengthen the statement that "This high rate of true positives (13/14=93%) suggests other untested, size-associated genes to be novel growth regulators."
3. Page 18 - should the reference to Fig 2B actually reference Fig 6B?
4. Fig 2F: The literature suggests *Stat92E* promotes growth. e.g. <https://www.ncbi.nlm.nih.gov/pmc/articles/PMC2914209/> Why does the association between *Stat92E* mRNA and size appear to be negative (ie higher levels associated with smaller wings) in Fig 2F? The authors should discuss this.
5. Likewise, in Suppl. Fig 3, higher *Tkv* expression is associated with smaller wings, but in the literature *Tkv* is thought to promote growth?
6. Can the authors discuss the advantages of looking at mRNA levels (as done here) compared to the

protein levels (already published), given that the two don't correlate, and protein levels are what matter at the end of the day? e.g. one can also see lowly-expressed genes, and non-coding RNAs?

1st Revision - authors' response

23rd September 2019

Responses to the reviewer's comments

Reviewer #1:

This is a very well written paper that is data dense. The topic fits the scope of the journal very well and would likely appeal to the readership. The paper uses the *Drosophila* genetic reference panel (DGRP) as a reference population to dissect the genetics of wing size using a systems biology approach. The data contain several layers. 1) DNA sequences collected by the DGRP; 2) gene expression of both sexes in 32 extreme lines in imaginal discs. 3) protein expression from a previous study in 30 lines; I believe not all of these lines overlap with the 32 RNA expression lines but it's not very clear from the paper. 4) wing size trait data from these selected lines and other lines, also collected from a previous study. Then the authors performed a series of analyses to correlate these different levels of variation and interpret their data using analysis such as pathway enrichment, etc. The analyses are sound and comprehensive. They found that canonical growth genes don't contribute much to the natural variation. Instead, they found a bunch of non-canonical genes that seem to regulate wing size variation within each sex. The most interesting and valuable part of the paper is the prediction piece, in which they are able to predict the phenotype of wing size by summing up allelic effects (+ or -) of a handful of SNPs that are associated with the trait and RNA expression. There are several main comments I have regarding the analyses and interpretation, which I think are important.

1) Figure 2E: The y axis of this figure is association P values for type 3. However, my understanding is that your type 3 associations were identified through type 1 analysis. So the P values are not for type 3 but type 1. In addition, I think it would make more sense to make a plot of the same x, and y but for all genes (without selecting canonical growth genes). If all genes have the same pattern, then it's not surprising that a selected subset also shares the same pattern. Based on Figure 2A, the majority of genes are type 3, I suspect this to be true. This is very important and affects some of the interpretations.

Response: The P values for type 3 association, plotted in the y axis of the Figure 2E, were evaluated through type 3 analysis (ANOVA on sex), not through type 1 analysis (regression on size trait). This is now clarified in the figure legend for Figure 2E.

As the reviewer points out, if we make the same plot using all the associated genes (without selecting canonical growth genes), the plot should make a similar pattern to Figure 2E. This is obvious since the majority of the size-associated genes are genes with type 1-exclusive association, as shown in Figure 2A, and the genes with type 1-exclusive association are mostly genes with type 3 association, as shown in Appendix Figure S1. Based on these results, we mentioned in page 8 (line 15) that a large part of the size-associated genes is involved in sexual size dimorphism. However, what is striking is that canonical growth genes followed this pattern even though we have long expected that canonical growth genes should show good associations with size through both type 2 and type 3. The finding is against our long-held thought that canonical genes are main regulators responsible for between-line size variation.

2) Figure 4A: the x-axis is SNP effect, but the y axis is minimum p value for a gene. Can you plot the P values of SNPs on the y axis instead? I don't see the necessity to use a gene minimum p value.

Response: Following the reviewer's suggestion, Figure 4A has been modified so that P values of all the eQTLs (SNPs) are plotted on the y axis. The top SNPs are now labeled with the corresponding gene names.

3) Figure 5A (now as Figure 5B): I don't understand why this figure allows you to display "mediators". Some graph to explain what your definition of mediators is would be helpful.

Response: We defined "mediators" as genes that exhibited triangular associations among SNP variation, the expression level and wing size phenotype. We identified mediator genes, first by selecting RNAs whose expression is doubly associated with both variations in SNPs and wing size traits (absCS/relCS) at 20% FDR, and second by selecting RNAs whose associated SNPs are also associated with wing size traits. We have now added a panel (new Figure 5A) that explains the definition and selection process of mediators. The main text also explains this (page 16-17).

4) One of the main conclusions of the paper is that a few trait associated SNPs are able to predict phenotypes with good accuracy. $R^2 = 0.16$ translates to correlation of 0.40, which is pretty good compared to genomic prediction in *Drosophila*. However, it's not clear that when prediction was made on 143 lines whether these lines include the 32 lines used in the present study. The correct analysis is to run this with an independent set of

lines. If the 32 lines were among the 143 lines, please remove them and re-evaluate the correlation between the predicted and observed wing sizes. This is very important because including the lines in the test set can significantly bias the prediction accuracy upwards.

Response: As the reviewer points out, our population-level prediction included the 32 lines. We checked predictability using the rest of 111 lines only without the 32 lines, and found the correlation with wing size was not good (Pearson correlation coefficient $\rho=0.056/0.041$ (P value= $0.28/0.33$) for female absolute/relative wing size predictions). This probably occurred because, without the 32 lines with extreme wing sizes, the wing size variation of the 111 lines was much smaller than the population-level size variation. Indeed, the majority of the wing sizes from the 111 lines is concentrated around the population mean as shown in Appendix Figure S10A).

We reasoned that the limited predictability came from the fact that we applied the same gene and SNP sets in the model to the 4 different biological situations (male/female, absolute/relative wing size), even though we identified different sets of mediator genes and SNPs for each situation. Another factor that possibly compromised the predictability is SNP numbers we included in the model. We only used the top-associated SNP for each mediator gene so that the model did not consider the relative importance of genes that may be represented by the number of associated SNPs.

We have reconstructed a modified model that now considers only genes relevant for each situation (sex/wing size trait), and consider all SNPs corresponding to the mediator genes in each situation (as selected in Figure 5B). The genes and SNPs used for each situation are described in Dataset EV8. Now the new model successfully predicts the wing sizes of the medium-ranged 111 lines at the Pearson correlation of 0.14 – 0.20 with statistical significance ($p<0.05$) for all situations (except a case of male and relative wing size, $p=0.078$) (as shown in Appendix Figure S10B). The model now reproduces wing size variation of the 32 extreme lines at the Pearson correlation of 0.58 – 0.72 (as shown in Figure 6B) for all 4 situations, and predicts the population-wide wing size (143 lines) at the Pearson correlation of 0.29 – 0.36 (as shown in Figure 6C).

The main text now describes the new model (page 18 and 19).

Minor comments:

1. Bottom of page 3: "it is fully unknown" -> "it is not fully known"

Response: Corrected as suggested.

2. Page 4: The contrast between

Response: We could not find the expression "The contrast between" in page 4 of the manuscript.

3. Page 7, Figure 1E: these RNAs have the smallest sd, do they also have the highest mean expression? I wonder what this means.

Response: As suggested by the reviewer, we have confirmed that the RNAs have the highest mean expressions (as shown in new Figure 1E). This suggests that the RNAs co-quantified at the protein level are mostly abundant RNAs that probably make them stable (invariable) among lines. Presumably, their functions are vital so that their expression cannot be highly variable among individuals in the population.

We now describe the analysis of the mean expression in the manuscript (page 7, lines 5-7)

Reviewer #2:

This is an interesting manuscript that attempts to analyze the genetic basis for natural wing size variation between *Drosophila* sexes and between different natural strains. Overall, the authors have done a good job of their analysis and presentation of the findings. I recommend publication.

Minor comments:

1. In the abstract, the authors should mention that the Ecdysone Receptor was identified as being differently expressed between fly lines of different wing size, along with other novel genes not previously implicated in size control. The EcR result is important and should be highlighted rather than be buried. It is interesting that size differences between mammals (including sex differences) are also strongly governed by steroid hormones.

Response: We have added the following sentence in the abstract: “Only a few growth genes including a receptor of steroid hormone ecdysone exhibit association with between-line size differences.”

We now have discussed EcR in the “Discussion” section of the manuscript (page 21-22) as follows: We should note that EcR, one of the canonical growth regulators, exceptionally exhibited a significant association with between-line wing size at $FDR < 20\%$. A molting hormone, ecdysone, and its nuclear receptor EcR have been thought to provide a substantial contribution to *Drosophila* body and organ size (Mirth and Shingleton, 2012). EcR forms a heterodimer with Ultraspiracle (*usp*) and, without ecdysone, binds to the promoter of target genes and represses transcription (Schubiger, 2005). Once EcR binds to ecdysone, the transcriptional repression by the unliganded EcR-*usp* complex is eliminated and EcR, recruiting transcriptional activators and co-factors, activates transcription, thus allowing developmental transitions to proceed. Knockdown of EcR has been shown to derepress transcription, which also allows developmental programs to proceed, mimicking ecdysone action (Mirth et al., 2009). We now observed the negative correlation of EcR levels with between-line wing size variation. A possible explanation would be that higher levels of EcR requires a smaller amount of ecdysone for developmental transitions through transcriptional activation and therefore, higher levels of EcR expedite the transitions, resulting in a shorter growth period and smaller tissues. It is also possible that the variability in ecdysone titers secreted from prothoracic glands and its temporal profiles affect between-line wing size non-autonomously. The relative contribution of EcR compared to that of novel regulators that were identified in the study remains to be determined.

2. Is it also worth mentioning that not a single component of the Insulin-IGF1 pathway was identified in this analysis, despite the fact that IGF1R is crucial in variation between species of dogs (artificially selected)? This might be because IGF1R-variation leads to large dogs with short life spans, while natural variation leads to large mammals with long life spans.

Response: We discuss this issue in the “Discussion” section of the manuscript (page 22): Insulin-like signaling and mTOR (insulin/TOR) pathway have been thought as the key players in the environmental and physiological control of body/organ size in *Drosophila* (Mirth & Shingleton, 2012). Our list of canonical growth genes includes several key components from the insulin/TOR pathway such as *Ilp2* (insulin-like peptide), *InR* (insulin-like receptor), *Pi3K92E* (catalytic subunit of phosphatidylinositol 3-kinase), *Pten* (phosphatase and tensin homolog), *Akt1* (protein kinase B) and *foxo* (forkhead box, sub-group O transcription factor). Among these, *InR* was identified to be associated with wing size at $FDR < 20\%$ (Fig 2E). The positive association to wing size was through type 3, suggesting its contribution to sexual size dimorphism. GWAS and eQTL mapping in our studies did not find any SNPs associated in the *InR* region. However, GWAS of domestic dogs that experienced intensive selection for size has found that alleles at insulin-like growth factor 1 (*IGF1*) and its receptor (*IGF1R*: mammalian ortholog of *InR*) are critical to the size variation (Sutter et al, 2007; Hoopes et al, 2012). This may indicate that very rare alleles that usually do not survive in natural settings were artificially selected in the domestic dog breeding.

3. Did the authors attempt to RNAi their HINT homolog in the wing? Is there any chance that the variation in HINT might also affect the neighbouring downstream gene *eIF2a*?

Response: We have found a single UAS-RNAi line for CG34015, the HINT homolog, available from Bloomington *Drosophila* stock center. We crossed this line with a hedgehog-Gal4 driver line, as we performed in Figure 3, and observed a significant reduction of size in the P-compartment of the wing in both sexes (as shown in Figure 5C and D). We have thus revealed that CG34015 positively regulates wing size. The experiment is now described in the main text (page 17, lines 11-13).

eIF-2alpha, immediately downstream of CG34015, shows a strong sexual dimorphic expression pattern at both RNA and protein levels (P value= 3×10^{-11} and 4×10^{-6} , respectively, for type 3 association), but does not exhibit between-line size association (P value=0.13 and 0.82 for type 2 association) as shown in Appendix Figure S11. Since RNA expression of CG34015 shows the strongest association only with between-line size differences (P value= 2×10^{-6} for type 2, 0.28 for type 3), the genetic control of *eIF-2alpha* should be unrelated to the one of CG34015. This is discussed in the “Discussion” section of the manuscript (page 23).

Reviewer #3:

A number of genetic studies in the past have identified genes that have the capability to affect tissue size during development if genetically perturbed. Importantly, however, this does not mean that these genes actually do regulate tissue size during development. They might simply be required for growth, but not be the regulatory node that the biological system uses to control growth physiologically. Hence an important open question is what genes control tissue and animal size in vivo in non-experimentally-perturbed conditions. This interesting study starts addressing this issue by looking at the association of mRNA levels in the developing *Drosophila* wing with final wing size, across fly lines and between males and females. The manuscript is well written. The data are interesting. For the most part, the conclusions are solid and well founded (with the two exceptions

mentioned below). The approach presented here seems to have been quite successful, identifying 13 new genes (out of 14 tested) that affect organ size. I think this study will be of interest to a wide range of people interested in development, organ size and growth, patterning, metabolic regulation, and signaling pathways that control cell growth and proliferation.

The following results would need to be strengthened:

1. The size phenotypes presented in Fig 3C would need to be confirmed/repeated with independent RNAi lines. Especially undergrowth phenotypes may reflect unhappy cells, and there is a good chance of them being due to off-target effects of the RNAis.

Response: To address the reproducibility of the RNAi results, we obtained two more independent RNAi lines for 9 genes (out of 14 tested) in addition to the lines we used for RNAi. For the rest of 5 genes, we could not find two more independent RNAi lines available in public but could obtain a single independent RNAi line for each of the genes. We crossed them with the hedgehog-Gal4 driver line and tested RNAi effect on the size of wing P-compartment. As shown in the new Figure 3C that now contains the results from the original and new experiments, all the 4 (negative) control cases resulted in the same range of P/A compartment size ratio, indicating a high reproducibility of the experiments. Consistent with the original result, new 2 cases of CG14207 knockdowns resulted in the same range of the P/A size ratio as the controls (even though the P value of one case claims a significant change), supporting the observation in the original experiment that CG14207 may not contribute to wing size variation. In contrast, all other, new RNAi cases on 13 genes exhibited significant size changes, supporting the original experiment. Knockdown of 10 genes (out of the 13) led to reduced P-compartment sizes in all RNAi cases (even though the strength of the effects varies for some genes), suggesting that these genes are positive growth regulators. However, knockdown of the other 3 genes (CG7173, Ppn and yip2) showed significant size changes in opposite directions, dependent on lines used.

To see the direction of effect associated with wing size variation in natural populations, we have plotted RNA/protein levels of 13 genes against wing size variation in DGRP lines (Figure EV4). The plots revealed that out of the 13 genes, all genes except Ppn are positively correlated with wing size, suggesting that Ppn only is a negative growth regulator and the others are positive regulators. This supports that the 10 genes with consistent size reductions in all RNAi cases function as positive regulators. The plots support CG7173 and yip2 to be positive regulators and Ppn to be negative regulators, but more detailed assessments are needed to determine the direction in the size regulation of the 3 genes.

2. The authors identify CG34015 as an important regulator of wing size, but this is not functionally tested. Does modulation of CG34015 expression (either RNAi or overexpression) affect wing size? Since the nearby SNPs are actually in the neighboring gene Rbp2, it would be good to also test this one.

Response: As described above (in the response to the comments from reviewer 2), a single line is available in public for CG34015 perturbation. We crossed the RNAi line with a hedgehog-Gal4 driver line, as we performed in Figure 3, and observed a significant reduction of size in the P-compartment of the wing in both sexes (as shown in Figure 5C and D). We have thus revealed that CG34015 positively regulates wing size. The experiment is now described in the main text (page 17, lines 11-13).

We have already tested Rbp2 functionally before, as described in our previous GWAS paper. The GWA study tested SNPs located around the Rbp2 gene region but did not identify any single SNPs that were associated with wing size traits. In the current study, eQTL mapping also did not find any SNPs cis-associated with Rbp2 expression at FDR < 20%. However, in the GWAS paper, in addition to the single-marker GWAS, we also performed gene-based GWAS using the sum of chi-squares VEGAS method (Liu *et al.* 2010), which evaluates P values considering all SNPs within each gene. The gene-based analysis identified Rbp2 within the top 20 associations and we functionally tested Rbp2 by RNAi. We knocked down the gene level in the whole wing using nubbin-Gal4 driver by crossing with two independent RNAi lines. Rbp2 was among genes with the best reductions in size (~10%) in both sexes. In the current study, the Rbp2 RNA level exhibited a weak, positive correlation with between-line wing size variation (P-value=0.05 for type-2 association) but the protein level rather showed a sexual dimorphic expression pattern (P-value=0.78 for type-2, 0.0055 for type-3 association) as shown in Appendix Figure S11. Considering the CG34015's strong association to between-line size variation, it seems that Rbp2 is under a different genetic control and rather contribute to sexual size dimorphism. This is discussed in the "Discussion" section of the manuscript (page 23).

Minor Issues

1. Fig 1F - it is not completely clear from the main text and figure legend how this analysis was done. Were RNA and protein levels correlated within one fly strain, thereby yielding a correlation coefficient, and this was done for all 32 fly strains? (In which case I would expect the total sum of correlation coefficients in the curve in Fig 1F to be 32. However, it looks like the distribution of roughly 1000 correlation coefficients is shown in 1F.)

So instead, was the correlation performed comparing RNA and protein for one gene across the 32 fly strains, thereby yielding a correlation coefficient for each gene?) This should be spelled out a bit more.

Response: The correlation analysis was performed comparing RNA and protein levels for each gene across 56 overlapping biological conditions (28 fly lines x 2 sexes). We revised the description in the manuscript as follows (page 7, lines 7-9): “We next examined the distribution of the gene-based correlation between the levels of RNAs and proteins by comparing both levels across the 56 biological conditions for each gene”.

2. It is not very clear from the text how the 20 size-associated genes to test were selected. For instance, *yip2* appears at position 392 on the Suppl. Table 3 list when sorted by FDR. It would be good to explain in more detail how these 20 genes were selected, because by showing they were not cherry-picked would strengthen the statement that “This high rate of true positives (13/14=93%) suggests other untested, size-associated genes to be novel growth regulators.”

Response: The 20 genes were picked without considering their (if any) putative functions. We considered 2 factors: (1) the genes were highly associated with wing size traits at either RNA or protein level. The selection was biased toward genes with type 2 association. (2) Availability of RNAi lines in public. The RNAi line for *yip2* was found available in the lab and was picked as a non-type 2 gene candidate. Note that we have never performed RNAi on *yip2* before the study.

3. Page 18 - should the reference to Fig 2B actually reference Fig 6B?

Response: Corrected.

4. Fig 2F: The literature suggests **Stat92E** promotes growth.

e.g. <https://www.ncbi.nlm.nih.gov/pmc/articles/PMC2914209/>

Why does the association between Stat92E mRNA and size appear to be negative (ie higher levels associated with smaller wings) in Fig 2F? The authors should discuss this.

Response: We discuss this in the “Discussion” section of the manuscript (page 20, line 6) as follows: It appears that RNA/protein level differences between sexes seem to agree with the direction of sexual size difference (female wing > male wing) for canonical growth genes whose effect on size is evidently known: For instance, positive growth regulators such as *dm* (Myc proto-oncogene ortholog) have higher levels in females that have larger wings, but negative growth regulators such as *Gap1* (a GAP protein for Ras) and *aos* (EGF antagonist) have lower levels in females. However, as the reviewer points out, pro-growth Stat92E has lower levels in females. These facts suggest that the final sexual size difference is determined by balancing the outcomes of system-wide interactions between positive and negative regulations controlled by many growth genes and pathways.

5. Likewise, in Suppl. Fig 3, higher **Tkv** expression is associated with smaller wings, but in the literature *Tkv* is thought to promote growth?

Response: The same discussion goes as above.

6. Can the authors discuss the advantages of looking at mRNA levels (as done here) compared to the protein levels (already published), given that the two don't correlate, and protein levels are what matter at the end of the day? e.g. one can also see lowly-expressed genes, and non-coding RNAs?

Response: We discuss in the “Discussion” section of the manuscript (page 24-25) as follows:

A recent technological advancement in omics field provides us with tools by which we can investigate on molecular pathways that bridge genetic variation and phenotypic variation. Protein is expected to be the final determinants of phenotypes and therefore, protein abundance is functionally more relevant than RNA abundance. Recent large-scale omics studies including the current study have demonstrated a relatively weak correlation between abundance of proteins and RNAs, which implicates distinct genetic architectures between the two molecular layers along the information paths from genotypes to phenotypes. Based on the central dogma of molecular biology, RNA is situated more closely to genomic information than protein. It can be, therefore, expected that genomic variants may more closely associate with RNA expression. Indeed, our study has shown concentrated eQTLs at the TSS and TES of transcripts, suggesting genetic controls of RNA metabolism. In the study, by adding the transcriptomic analysis to the previous proteomics analysis, our data add information on non-coding genes and lowly-expressed genes and now provide a more comprehensive, detailed view on the genetic information paths from genotypes through molecular layers to size phenotypes.

Thank you for sending us your revised manuscript. We have now heard back from two of the three reviewers who were asked to evaluate your study. Since their recommendations are quite similar, I prefer to make a decision now rather than further delaying the process. The reviewers are now overall supportive. I am pleased to inform you that your manuscript will be accepted in principle pending the following essential amendments.

REFEREE REPORTS

Reviewer #2:

The authors have responded very well to all three reviewers. I recommend publication without further delay.

Reviewer #3:

The authors have addressed the issues raised in my original review.

Corresponding Author Name: Hirokazu Okada

Manuscript Number: MSB-19-9012